# Global Evaluation of Gross Primary Productivity in the JULES Land Surface Model v3.4.1

Darren Slevin[1], Simon F. B. Tett[1], Jean-François Exbrayat[1, 2], A. Anthony Bloom[1, 2, 3], and Mathew Williams[1, 2]

[1]School of GeoSciences, The University of Edinburgh, Crew Building, Alexander Crum Brown Road, Edinburgh, EH9 3FF, UK
[2]National Centre for Earth Observation, The University of Edinburgh, Crew Building, Alexander Crum Brown Road, Edinburgh, EH9 3FF, UK
[3]Jet Propulsion Laboratory, California Institute of Technology, Pasadena, CA 91109, USA

*Correspondence to:* Darren Slevin (d.slevin@ed.ac.uk)

**Abstract.** This study evaluates the ability of the JULES Land Surface Model (LSM) to simulate Gross Primary Productivity (GPP) at regional and global scales for 2001–2010. Model simulations, performed at various spatial resolutions and driven with a variety of meteorological datasets (WFDEI-GPCC, WFDEI-CRU and PRINCETON), were compared to the MODIS GPP product, spatially gridded estimates of upscaled GPP from the FLUXNET network (FLUXNET-MTE) and the CARDAMOM terrestrial carbon cycle analysis. Firstly, when JULES was driven with the WFDEI-GPCC dataset (at $0.5° \times 0.5°$ spatial resolution), the annual average global GPP simulated by JULES for 2001–2010 was higher than the observation-based estimates (MODIS and FLUXNET-MTE), by 25 % and 8 %, respectively, and CARDAMOM estimates by 23 %. JULES was able to simulate the standard deviation of monthly GPP fluxes compared to CARDAMOM and the observation-based estimates at global scales. Secondly, GPP simulated by JULES for various biomes (forests, grasslands and shrubs) at global and regional scales were compared. Differences among JULES, MODIS, FLUXNET-MTE and CARDAMOM at global scales were due to differences in simulated GPP in the tropics. Thirdly, it was shown that spatial resolution ($0.5° \times 0.5°$, $1° \times 1°$ and $2° \times 2°$) had little impact on simulated GPP at these large scales with global GPP ranging from 140–142 $\mathrm{PgC\,year^{-1}}$. Finally, the sensitivity of JULES to meteorological driving data, a major source of model uncertainty, was examined. Estimates of annual average global GPP were higher when JULES was driven with the PRINCETON meteorological dataset than when driven with the WFDEI-GPCC dataset by 3 $\mathrm{PgC\,year^{-1}}$. At regional scales, differences between two were observed with the WFDEI-GPCC driven model simulations estimating higher GPP in the tropics (5°N–5°S) and the PRINCETON driven model simulations estimating higher GPP in the extratropics (30°N–60°N).

## 1 Introduction

The land surface is an important component of the climate system, provides the lower boundary for the atmosphere and exchanges energy, water and carbon (C) with the atmosphere (Pielke et al., 1998; Pitman, 2003; Seneviratne and Stöckli, 2008). It also controls the partitioning of available energy (into latent and sensible heat) and water (into evaporation and runoff) at

the surface (Bonan, 2008). Changes in the land surface due to human activities, such as those from tropical deforestation, can influence climate at various time and spatial scales and since the land surface is the location of the terrestrial C cycle, its ability to act as a C source or sink can influence atmospheric $CO_2$ concentrations (Le Quéré et al., 2009; Pan et al., 2011; Le Quéré et al., 2013; Tian et al., 2016). The reduced ability of the land surface to absorb increased anthropogenic $CO_2$ emissions

in the future has been shown by models and inferred from observations (Friedlingstein et al., 2006; Canadell et al., 2007; Friedlingstein et al., 2014; Sitch et al., 2015). Friedlingstein et al. (2006) and Friedlingstein et al. (2014) have suggested that a major source of model uncertainty is the land C cycle which can affect the ability of earth system models (ESMs; also known as coupled carbon-cycle–climate models) to reliably simulate future atmospheric $CO_2$ concentrations and climate (Dalmonech et al., 2014).

Plants fix $CO_2$ as organic compounds through photosynthesis at the leaf scale and Gross Primary Productivity (GPP) is the total amount of C used in photosynthesis by plants at the ecosystem level (Beer et al., 2010; Chapin III et al., 2012). Photosynthesis at the leaf and canopy scale vary in response to changes in climate (temperature, precipitation, humidity and downward radiation fluxes) and nutrient availability (Anav et al., 2015). Terrestrial GPP is an important (and the largest) C flux since it drives several ecosystem functions such as respiration and growth (Beer et al., 2010). GPP contributes to the production

of food, fibre, and wood for humans and along with respiration, is one of the major processes controlling the exchange of $CO_2$ between the land and atmosphere (Beer et al., 2010). It also plays an important role in the global C cycle helping terrestrial ecosystems to partially offset anthropogenic $CO_2$ emissions (Janssens et al., 2003; Cox and Jones, 2008; Battin et al., 2009; Anav et al., 2015)

However, at the global scale, there are no direct measurements of GPP (Anav et al., 2015). Global estimates of GPP exist, but

are not solely based on measurements and, therefore, large uncertainties exist in these estimates (Anav et al., 2015). In LSMs, the correct simulation of GPP is important since errors in its calculation can propagate through the model and affect biomass and other flux calculations, such as Net Ecosystem Exchange (NEE) (Schaefer et al., 2012). The correct representation of leaf level stomatal conductance influences GPP and transpiration and errors in GPP can also introduce errors into simulated latent and sensible heat fluxes.

Land surface models (LSMs) have become considerably more complex since the simple "bucket" model of Manabe (1969). Deardorff (1978) developed a model which could simulate temperature and moisture for two soil layers and included a vegetation layer. Sellers et al. (1986) built on the work of Deardorff (1978) by developing a globally applicable LSM. Foley et al. (1996) incorporated vegetation dynamics into an LSM. These developments have led to LSMs which can realistically represent complex vegetation responses to meteorology, the climate effect of snow and biogeochemical processes (Pitman, 2003; van den

Hurk et al., 2011). Therefore, as LSMs become more complex, their accuracy must be evaluated.

JULES has been evaluated at various scales: point (Blyth et al., 2010, 2011; Slevin et al., 2015; Ménard et al., 2015), regional (Galbraith et al., 2010; Burke et al., 2013; Chadburn et al., 2015) and globally as part of model-intercomparison studies (Anav et al., 2015; Sitch et al., 2015). Evaluating simulated GPP at a range of scales and its sensitivity to spatial resolution and meteorological data is essential for informing future model developments. In this manuscript, we do this using

two observation-based datasets (FLUXNET-MTE and MODIS) and the CARbon DAta MOdel fraMework (Bloom et al., 2016, CARDAMOM).

In this study, the ability of JULES version 3.4.1 to simulate global and regional fluxes of GPP for various biomes, spatial resolutions and using different meteorological data to drive the model is evaluated. In particular, the following research questions are addressed:

– How do estimates of global GPP compare to those from the observation-based datasets and the CARDAMOM framework? Can JULES capture interannual variability of GPP at the global scale?

– How does JULES GPP compare for various biomes at the global and regional scales?

– How sensitive are fluxes of GPP to the spatial resolution of the model?

– Is the meteorological data set used to drive the model important at the global scale?

## 2 Methods and model

### 2.1 Model description

The Joint UK Land Environment Simulator (JULES) is the land surface scheme of the UK Met Office Unified Model (MetUM), which is a single model family used to simulate weather and climate across a range of timescales (Walters et al., 2016). JULES is a community land surface model which has evolved from the Met Office Surface Exchange Scheme (MOSES) (Cox et al., 1999) and is used for modelling all of the processes at the land surface, in the sub-surface soil, and surface exchange processes (Best et al., 2011; Clark et al., 2011). JULES can be used *offline* (i.e. outside of the host ESM, MetUM) and model simulations can be performed at point, regional and global scales. Plant Functional Types (PFTs) are used to represent broad groupings of plant species with similar ecosystem functions and resource use. In the version of JULES used in this study (version 3.4.1), each model gridbox consists of 9 different surface types; 5 PFTs (broadleaf trees, needleleaf trees, C3 (temperate) grass, C4 (tropical) grass and shrubs) and 4 non-vegetation surface types (urban, inland water, bare soil and land-ice). Model gridboxes can consist entirely of a mixture of the first 8 surface types or only land-ice. Since model version 4.2, each JULES gridbox can contain nine PFTS (tropical broadleaf evergreen, temperate broadleaf evergreen, broadleaf deciduous, needleleaf evergreen, needleleaf deciduous, C3, C4, evergreen shrub, deciduous shrub) (Harper et al., 2016).

JULES is driven by the downward shortwave and longwave radiation fluxes, rainfall and snowfall rates, surface air temperature, wind speed, surface pressure and specific humidity. The downward shortwave and longwave radiation fluxes play an important role in the surface energy balance, where the downwelling radiation fluxes must equal the outgoing fluxes of sensible heat, latent heat, ground flux, reflected shortwave radiation and upwelling thermal energy, and the calculation of photosynthesis (Best et al., 2011; Clark et al., 2011). GPP is the total C uptake by plants in photosynthesis at the canopy scale with potential (without water and ozone stress) leaf-level photosynthesis calculated as the smoothed minimum of three limiting rates: (1) Rubisco-limited rate (determined using surface air temperature and atmospheric $CO_2$ concentrations), (2) Light-limited rate

(determined using downward shortwave radiation) and (3) Rate of transport of photosynthetic products ($C_3$ plants) and PEP-Carboxylase limitation ($C_4$ plants) (determined using surface air temperature and pressure) (Clark et al., 2011). Soil moisture stress is taken into account when calculating leaf-level photosynthesis by multiplying the potential leaf-level photosynthesis by a soil moisture factor (determined using mean soil moisture concentration in the root zone).

In JULES, there are two options available for radiation interception and the scaling of photosynthesis from leaf-level to canopy-level: (i) big leaf approach and (ii) multi-layer approach. For all model simulations performed in this study, the multi-layer approach was used which takes into account the vertical gradient of canopy photosynthetic capacity (decreasing leaf nitrogen from top to bottom of canopy) and includes light inhibition of leaf respiration (Option 4 in Table 3 of Clark et al. (2011)). Canopy-scale fluxes are estimated to be the sum of the leaf-level fluxes in each canopy layer, scaled by leaf area. LAI
is calculated for each canopy level (default number is 10), with a maximum LAI prescribed for each PFT.

Phenology (bud burst and leaf senescence) in JULES is usually updated once per day by multiplying the annual maximum LAI by a scaling factor (calculated using accumulated temperature-dependent leaf turnover rates). For each PFT, the C fluxes are calculated using a coupled photosynthesis-stomatal conductance model on each model timestep (typically 30 to 60 minutes) (Cox et al., 1998). These fluxes are then time-averaged (usually every 10 days) before being passed to TRIFFID (Top-down
Representation of Interactive Foliage and Flora Including Dynamics), JULES' dynamic global vegetation model, which updates the vegetation distribution, based on the net C available to it and competition with other vegetation types, and soil C in each model gridbox on a longer timestep (usually every 10 days) (Cox, 2001). Clark et al. (2011) and Best et al. (2011) contain a more detailed description of JULES.

## 2.2   Experimental design

Offline simulations of GPP were performed at the global scale for the 2001–2010 period using various meteorological datasets and spatial resolutions (Table 1). A general overview is provided of how sensitive JULES GPP is to the meteorological dataset used at global scales rather than for each meteorological variable. By analysing the models sensitivity to each meteorological dataset, different analyses of the global climate are compared and therefore a multi-factor analysis of combined changes in meteorological variables can be performed. The land cover was kept constant at values for the year 2000 (Loveland et al.,
2000) and annual atmospheric $CO_2$ concentrations were varied as in the historical record. The 2001–2010 time period was used to due to the availability of multiple global meteorological and GPP datasets for this period. JULES is compared against FLUXNET-MTE, MODIS and CARDAMOM GPP.

Prior to performing the global scale model simulations, the soil moisture was brought to equilibrium using a 40 year global spin-up by cycling 10 years of meteorological data (1979–1989) twice and 10 years of meteorological data (1989–1999)
twice (in equilibrium mode), followed by a 12 year spin-up by cycling 12 years of meteorological data (1999–2010) once (in dynamical mode). Clark et al. (2011) contains more information on spinning up the soil C pools.

## 2.3   Data

The datasets used in this study include those used as input to JULES (soil, vegetation and meteorological data) and the bench-marking data. The soil dataset used was the Harmonized World Soil Database version 1.2 (Nachtergaele et al., 2012, HWSD) and contains soil property data such as soil texture fractions, water storage capacity, soil depth and pH (Nachtergaele et al., 2012). In this study, the soil texture fractions (% of sand, silt and clay) were used to calculate the soil thermal and hydraulic conductivity parameters listed in Table 3 of Best et al. (2011). The land cover classification scheme used for specifying the PFT fractions for each model gridbox at the global scale was Global Land Cover Characterization database version 2.0 (Loveland et al., 2000, http://edc2.usgs.gov/glcc/glcc.php). Two meteorological datasets were used to drive the model offline (i.e. run separately from its host Earth System Model) at global scales; WFDEI (Weedon et al., 2014) and PRINCETON (Sheffield et al., 2006).

Global gridded estimates of GPP derived from the upscaling of observations from the FLUXNET network of tower sites (Jung et al., 2009), estimates from the Moderate Resolution Imaging Spectroradiometer (MODIS) sensor, aboard the U.S. National Aeronautics and Space Administration (NASA) Earth Observation System (EOS) satellites, Terra and Aqua (Yang et al., 2006), and GPP simulated by the CARbon DAta MOdel fraMework (Bloom et al., 2016, CARDAMOM) were used to evaluate model performance. These global gridded estimates of GPP provide a means to evaluate LSMs at large scales (Jung et al., 2009, 2010; Beer et al., 2010; Zhao and Running, 2010; Bonan et al., 2011; Lei et al., 2014).

### 2.3.1   Forcing data

As part of the EMBRACE EU FP7 programme (http://www.embrace-project.eu/), the WATCH Forcing Data (WFD) methodology was applied to the ERA-Interim reanalysis data for the 1979–2013 period to generate the WFDEI meteorological forcing data (Weedon et al., 2014). As for the WFD, WFDEI has two precipitation products, corrected using either CRU (Climate Research Unit at the University of East Anglia) or GPCC (Global Precipitation Climatology Centre) precipitation totals (Weedon et al., 2014) and are referred to as WFDEI-CRU and WFDEI-GPCC, respectively. The GPCC data product is a gridded gauged precipitation dataset and provides a higher resolution dataset (i.e. better station coverage, particularly at high latitudes, and especially for the end of the 20[th] century) than the CRU precipitation totals (Weedon et al., 2014). The WFDEI dataset consists of 3 hourly, regularly gridded data at half-degree ($0.5° \times 0.5°$) spatial resolution and is only available for land points including Antarctica. The dataset contains the following meteorological variables: downward shortwave and longwave radiation fluxes ($\mathrm{W\,m^{-2}}$), rainfall rate ($\mathrm{kg\,m^{-2}\,s^{-1}}$), snowfall rate ($\mathrm{kg\,m^{-2}\,s^{-1}}$), 2 m temperature (K), 10 m wind speed ($\mathrm{m\,s^{-1}}$), surface pressure (Pa) and 2 m specific humidity ($\mathrm{kg\,kg^{-1}}$).

The PRINCETON dataset is a global 62 year near-surface meteorological data set used for driving land surface models and was created by Princeton University's Terrestrial Hydrology Group (Sheffield et al., 2006, http://hydrology.princeton.edu/home.php). The PRINCETON data set consists of 3 hourly, regularly gridded data at 1-degree ($1° \times 1°$) spatial resolution for the 1948–2010 period and is only available for land points excluding Antarctica. The dataset contains the same meteorological variables as WFDEI with the exception of rainfall and snowfall rates summed as total precipitation ($\mathrm{kg\,m^{-2}\,s^{-1}}$).

### 2.3.2 Benchmarking data

The upscaled FLUXNET GPP (hereafter referred to as FLUXNET-MTE) was derived using a model tree ensemble (MTE) approach, a type of machine learning technique that can be trained to predict land-atmosphere fluxes (Jung et al., 2009). Based on observed meteorological data, land cover data and remotely sensed vegetation properties (fraction of absorbed photosynthetic active radiation), the upscaling principle can predict estimates of C fluxes at FLUXNET sites with available quality-filtered flux data and the trained model is then applied spatially using grids of the input data (Jung et al., 2009, 2011). However, these machine learning algorithms are typically data limited due to the quantity, quality and representativeness of the training dataset (Jung et al., 2009). There are two upscaled FLUXNET GPP datasets available depending on the flux partitioning method used to separate net ecosystem exchange of $CO_2$ (NEE) into GPP and terrestrial ecosystem respiration (TER) (Reichstein et al., 2005; Lasslop et al., 2010). In this study, GPP based on the work by Reichstein et al. (2005) was used (this is the flux partitioning method used by the FLUXNET network). However, differences between the two upscaled FLUXNET GPP datasets are small. FLUXNET-MTE is a $0.5° \times 0.5°$ spatial and monthly temporal resolution data set for the 1982–2011 period and is available from the Max Planck Institute for Biogeochemistry Data Portal (https://www.bgc-jena.mpg.de/geodb/projects/Home.php).

The MOD17 MODIS Gross/Net Primary Productivity (GPP/NPP) product provides continuous estimates of GPP/NPP for the Earth's entire land surface and is produced as part of the NASA's Earth Observing System (EOS) program. The MOD17 algorithm produces two subproducts, MOD17A2 (which stores 8-day composite GPP, net photosynthesis and QC flags) and MOD17A3 (annual NPP and QC flags) (Zhao et al., 2005). The resulting datasets contain regular gridded global estimates of GPP and NPP for the terrestrial land surface at the 1 km spatial resolution (Running et al., 2000). The Numerical Terradynamic Simulation Group (NTSG) (http://www.ntsg.umt.edu/project/mod17) at the University of Montana corrected problems associated with GPP estimates by spatial interpolation of the coarse resolution meteorological data, temporal infilling of cloud-contaminated MOD15A2 LAI/FPAR data and modification of BPLUT (Biome Property Look-Up Table) parameters based on observed GPP from flux tower measurements in order to create an improved MOD17 GPP product (Zhao et al., 2005). The global monthly MODIS GPP (version 55) dataset at $0.05° \times 0.05°$ spatial resolution for the 2001–2010 period was downloaded from (ftp://ftp.ntsg.umt.edu/pub/MODIS/NTSG_Products/). For the purposes of this study, the data was regridded to $0.5° \times 0.5°$ spatial resolution using the first order conservative remapping function (remapcon) of the Climate Data Operators (CDO) software package (https://code.zmaw.de/projects/cdo).

The CARbon DAta MOdel fraMework (CARDAMOM) is a model-data fusion approach which consists of merging observational data with models in order to improve model quality and characterise its uncertainty (Bloom and Williams, 2015; Bloom et al., 2016). CARDAMOM relies on a Bayesian Markov Chain Monte Carlo (MCMC) algorithm to explore the parametric uncertainty of the ecosystem C balance model Data Assimilation Linked Ecosystem Carbon Model version two (Bloom et al., 2016, DALEC2) according to available C relevant data-streams (fluxes, leaf area index, changes in biomass, etc.). CARDAMOM can be applied at the point-scale and spatially with available remote-sensing based products such as MODIS LAI, biomass and soil carbon maps. When the framework is applied spatially, the Bayesian model-data fusion approach is performed in every model gridbox independently without using pre-defined biome maps. C fluxes, pool increments and parameter values

with explicit confidence intervals attached to them are output from the MCMC algorithm. In this study, MODIS LAI, a tropical biomass map (Saatchi et al., 2011), a soil C dataset (Hiederer and Köchy, 2011), MODIS burned area (Giglio et al., 2013) and the ERA-Interim reanalysis data have been used as input to CARDAMOM in order to produce a global monthly mean GPP dataset at $1° \times 1°$ spatial resolution for the 2001–2010 period (Bloom et al., 2016).

## 2.4 Outline of experiments

This section describes the model simulations performed in this study (Table 1). For the JULES model simulations, the first part of the model simulation name refers to JULES version 3.4.1 and the second part refers to the global gridded meteorological dataset used to drive the model (Table 1). The spatial resolution of the model grid is appended to the end of the model simulation name. Model simulation names without an attached spatial resolution mean that the model simulation was performed at $0.5° \times 0.5°$ spatial resolution. Vegetation competition (simulated by TRIFFID, JULES' dynamic global vegetation model) has been switched off for the majority of model simulations. This was done in order to prevent unrealistic vegetation fractions in model gridboxes for global scale simulations of GPP. Differences between having prescribed PFTS (no vegetation competition) and allowing competition between PFTs was also examined. For the CARDAMOM simulation, the ERA-Interim reanalysis product was used to drive the DALEC2 model at $1° \times 1°$ resolution. Model results were compared to FLUXNET-MTE, MODIS and CARDAMOM GPP.

Firstly, model estimates of total annual GPP (JULES-WFDEI-GPCC) were integrated globally. The ability of JULES to simulate the interannual variability (IAV) of GPP at global scales was examined from 2001–2010 (JULES-WFDEI-GPCC; Table 1). Secondly, the modelled and observation-based estimates of GPP were further compared by biome type (Forest, Grassland and Shrub) at global and regional scales (Global, Tropics and Extratropics). The Forest, Grassland and Shrub biomes were determined by summing the PFT fractions in the land cover map for the broadleaf and needleleaf tree surface types, the C3 and C4 surface types and the shrub surface type, respectively, and dividing each by the sum of the fractions of the 5 PFTs in order to exclude the non-vegetation land cover types. GPP was analysed by biome type at regional scales by dividing the global land area into seven regions (Figure 1; Table 2). Thirdly, the sensitivity of the model to the spatial resolution of the input data was evaluated by varying the resolution of the ancillary data (soil and vegetation) and meteorological data (WFDEI-GPCC) (Table 1). The input data was regridded from $0.5° \times 0.5°$ to $1° \times 1°$ spatial resolution (JULES-WFDEI-GPCC-1degree; Table 1) and from $0.5° \times 0.5°$ to $2° \times 2°$ spatial resolution (JULES-WFDEI-GPCC-2degree) using the Climate Data Operators (CDO) software package. Further information on how the datasets were regridded can be found in Appendix D of Slevin (2016).

Finally, the sensitivity of JULES to the meteorological driving data was evaluated by comparing model simulations driven using the WFDEI-GPCC (JULES-WFDEI-GPCC-1degree; Table 1) and PRINCETON datasets (JULES-PRINCETON; Table 1) at $1° \times 1°$ spatial resolution (the same soil and vegetation ancillary datasets were used by both). The model's sensitivity to precipitation was examined by driving it with the WFDEI-CRU dataset (JULES-WFDEI-CRU; Table 1) at $0.5° \times 0.5°$ spatial resolution.

*2.5 Model analyses*

In order to quantify how the model performs at the global scale, the following metrics were used: global area-weighted mean ($\bar{x}$; Equation 1), Coefficient of Variation (CV; Equation 2) and monthly anomalies (Equation 3).

$$\bar{x} = \frac{\sum_{i,j=1}^{i=m,j=n} a_{i,j}\, x_{i,j}}{\sum_{i,j=1}^{i=m,j=n} a_{i,j}} \tag{1}$$

The global area-weighted mean is calculated by multiplying the monthly GPP flux for each grid box ($x_{i,j}$) by the area of its grid box ($a_{i,j}$) and dividing the sum of these values by the total land surface area. $m$ and $n$ are the total number of grid boxes in the x- and y-direction, respectively. For example, when running a global scale model simulation at half-degree ($0.5° \times 0.5°$) spatial resolution, $m = 720$ (number of grid boxes in the west-east direction) and $n = 360$ (number of grid boxes in the north-south direction).

$$\mathrm{CV} = \frac{\sigma}{\mu} \times 100 \tag{2}$$

CV (also known as relative variability) is a measure of the relative magnitude of the standard deviation ($\sigma$) and is calculated by dividing the standard deviation by the mean ($\mu$). It is expressed as a percentage and is always positive. CV is a useful statistic since it allows the degree of variation of various datasets to be compared even if the means are quite different from each other. It is also dimensionless which means that CVs can be used to compare the dispersion (variability) of the data when

other measures like standard deviation or root mean squared error cannot.

To quantify model performance at the global scale, CV was calculated by first computing the standard deviation and means of the global area-weighted means for each month and then dividing the average of the standard deviations by the average of the means for each month.

Monthly anomaly $= x - \bar{x}_{clim}$ $\tag{3}$

The monthly anomaly is defined as the departure of the observed monthly values ($x$) from the long-term (climatological) average for that month ($\bar{x}_{clim}$).

## 3  Results

*3.1  Global GPP*

In general, JULES simulates higher annual average global GPP than MODIS, FLUXNET-MTE and CARDAMOM with JULES
GPP closer to FLUXNET-MTE estimates. When driven with the WFDEI-GPCC dataset (JULES-WFDEI-GPCC; Table 1), JULES simulates global GPP with an annual average of $140\,\mathrm{PgC\,year^{-1}}$ (the combined GPP of all terrestrial ecosystems) over the 2001–2010 period (Figure 2c). This value is greater than that estimated by MODIS, FLUXNET-MTE and CARDAMOM

with annual average global GPP estimated to be 112, 130 and 114 $PgC\,year^{-1}$, respectively, for the same period (Figures 2a, b and d). The higher global GPP simulated by the JULES-WFDEI-GPCC driven simulations is greater than the MODIS, FLUXNET-MTE and CARDAMOM estimates by 25 %, 8 % and 23 % on average, respectively.

The difference in average annual global GPP between JULES-WFDEI-GPCC and MODIS (both at $0.5° \times 0.5°$ spatial resolu-
tion) is greater ($28\,PgC\,year^{-1}$) than that between JULES-WFDEI-GPCC and FLUXNET-MTE ($10\,PgC\,year^{-1}$) and between JULES-WFDEI-GPCC and CARDAMOM ($26\,PgC\,year^{-1}$). This difference between the model simulated and observation-based GPP estimates is also shown in the zonal mean of the total annual JULES-WFDEI-GPCC, MODIS, FLUXNET-MTE and CARDAMOM GPP with the largest differences between datasets found in the tropics at 10°S-10°N and subtropics at 15°N-30°N (Figure 2e).

*3.2   Seasonal and interannual variability of GPP*

Overall, it was found that JULES can simulate seasonal and interannual variability of GPP at global scales. JULES simulates the seasonal cycle of GPP (JULES-WFDEI-GPCC; Table 1) (Figure 3a) with the global average of its monthly GPP for 2001–2010 falling within range of the observation-based estimates (FLUXNET-MTE and MODIS) for much of the year (between 64 and $107\,g\,C\,m^{-2}\,month^{-1}$). A similar trend can be found with the CARDAMOM GPP (Figure 3a). The exception to
this are the Northern Hemisphere winter months (January, February, March and December) with JULES simulating higher global mean GPP by $2\,g\,C\,m^{-2}\,month^{-1}$ on average compared to FLUXNET-MTE. The MODIS GPP means are lower than FLUXNET-MTE for each of the monthly climatologies by $10\,g\,C\,m^{-2}\,month^{-1}$ on average (Figure 3a).

The standard deviation of the monthly GPP fluxes is used to measure interannual variability and this is expressed as a percentage of the mean monthly GPP fluxes using coefficient of variation (CV). Low values of CV mean that differences
between the monthly GPP fluxes and the mean monthly GPP fluxes are small and larger CV values mean the opposite. The CV of the model simulated and observation-based GPP fluxes range between 0.8–4 % for the mean monthly GPP with the highest differences between the monthly values being for Northern Hemisphere winter and spring (February, March, April, November and December) (Figure 3b). This pattern is similar to the global average of the monthly climatologies (Figure 3a).

The monthly anomalies (computed using the global mean values) expressed as percentages of the global mean of model
simulated monthly GPP (JULES-WFDEI-GPCC) compare equally well to both FLUXNET-MTE and MODIS GPP for 2001–2010 with both having Root Mean Squared Errors (RMSEs) of 2.4 % with CARDAMOM having much lower year to year variation (Figure 3c). JULES is able to capture simulated monthly anomalies from 2001 to 2010 with the exception of those in 2002 (Figure 3c).

*3.3   Global and regional comparison of simulated GPP for various biomes*

In addition to examining the ability of JULES to simulate global GPP (integrated across all ecosystem types), the total annual GPP for 2001–2010 was compared for various biomes (forests, grasslands and shrubs) at global and regional scales (Figure 4). This means that areas for model improvement can be identified at scales smaller than the global. JULES overestimates GPP

in all tropical land areas (Central and South America, Africa and South and South-East Asia), but is able to simulate it in the extratropics (Europe, Northern Asia, North America and Greenland and the Extratropical Southern Hemisphere) (Figure 4).

When JULES was driven with WFDEI-GPCC (JULES-WFDEI-GPCC), JULES simulated average annual GPP to be 68, 62 and $9\,\mathrm{PgC\,year^{-1}}$ for forests, grasslands and shrubs, respectively (Figure 4a). With the exception of shrubs, JULES over-estimates average annual GPP by 30 % (24 %), 12 % (7 %) and 21 % (28 %) compared to MODIS, FLUXNET-MTE and CARDAMOM GPP, respectively, for forests (grasslands) compared to MODIS, FLUXNET-MTE and CARDAMOM GPP (Figure 4a). Differences between JULES, MODIS, FLUXNET-MTE and CARDAMOM GPP for shrubs are small with average annual GPP ranging within $9$–$10\,\mathrm{PgC\,year^{-1}}$ (Figure 4a).

The differences in total annual GPP at the global scale is mainly due to differences between JULES and the observation-based (MODIS and FLUXNET-MTE) and CARDAMOM estimates in the tropics ($30°$S–$30°$N) (Figure 4b) with a large negative bias in JULES occuring in the subtropics at $15°$N-$30°$N (Figure 6). In the tropics, JULES simulates total annual GPP to be $55\,\mathrm{PgC\,year^{-1}}$, $44\,\mathrm{PgC\,year^{-1}}$ and $6\,\mathrm{PgC\,year^{-1}}$ for forests, grasslands and shrubs, respectively, for the 2001–2010 period. JULES overestimates total annual GPP by 19–40 % compared to MODIS, FLUXNET-MTE and CARDAMOM GPP for forests and by 22–52 % for grasslands in the tropical regions (Figure 4b). Differences between model simulated and observation-based estimates of GPP are small in the tropics for shrubs with total annual GPP ranging from $5$–$6\,\mathrm{PgC\,year^{-1}}$ (Figure 4b). In the extratropics ($30°$N–$90°$N and $30°$S–$90°$S), differences between model and observed GPP are small with average annual GPP for forests, grasslands and shrubs found to be $13$–$16\,\mathrm{PgC\,year^{-1}}$, $18$–$23\,\mathrm{PgC\,year^{-1}}$ and $3$–$5\,\mathrm{PgC\,year^{-1}}$, respectively (Figure 4c).

Total annual GPP at the regional scale was further examined by splitting the land area into seven regions (Figure 1; Table 2). The tropical regions ($30°$S–$30°$N) were divided up into three regions; Central and South America, Africa and South and South-East Asia. The extratropics ($30°$N–$90°$N and $30°$S–$90°$S) were divided into four regions; Europe, Northern Asia, North America and Greenland and the extratropical Southern Hemisphere. JULES overestimates GPP in all three tropical land areas compared to MODIS, FLUXNET-MTE and CARDAMOM (Figures 5c, e and f).

Differences in average annual GPP between JULES, MODIS, FLUXNET-MTE and CARDAMOM GPP range from $2.8$–$5.04\,\mathrm{PgC\,year^{-1}}$, $3$–$6.1\,\mathrm{PgC\,year^{-1}}$ and $0.8$–$3.7\,\mathrm{PgC\,year^{-1}}$ for forests, grasslands and shrubs, respectively, in South and South-East Asia, $3.5$–$4.6\,\mathrm{PgC\,year^{-1}}$, $3.2$–$4.2\,\mathrm{PgC\,year^{-1}}$ and $0.1$–$4.2\,\mathrm{PgC\,year^{-1}}$ for forests, grasslands and shrubs, respectively, in Africa and $1.9$-$5.6\,\mathrm{PgC\,year^{-1}}$, $1.7$–$4.5\,\mathrm{PgC\,year^{-1}}$ and $0.07$–$0.3\,\mathrm{PgC\,year^{-1}}$ for forests, grasslands and shrubs, respectively, in Central and South America (Figures 5c, e and f, respectively). In the extratropics, differences between JULES, MODIS, FLUXNET-MTE and CARDAMOM GPP are small with average annual GPP ranging from $0.08$–$0.5\,\mathrm{PgC\,year^{-1}}$, $0.1$–$1.5\,\mathrm{PgC\,year^{-1}}$ and $0.029$–$0.12\,\mathrm{PgC\,year^{-1}}$ for forests, grasslands and shrubs, respectively, in Europe, $0.09$–$1.4\,\mathrm{PgC\,year^{-1}}$, $0.7$–$1.8\,\mathrm{PgC\,year^{-1}}$ and $0.2$–$0.6\,\mathrm{PgC\,year^{-1}}$ for forests, grasslands and shrubs, respectively, in Northern Asia, $0.09$–$0.14\,\mathrm{PgC}$, $0.7$–$1.8\,\mathrm{PgC\,year^{-1}}$ and $0.03$–$0.2\,\mathrm{PgC\,year^{-1}}$ for forests, grasslands and shrubs, respectively, in the Extratropical Southern Hemisphere and $0.005$–$5\,\mathrm{PgC\,year^{-1}}$, $3$–$5\,\mathrm{PgC\,year^{-1}}$ and $0.3$–$0.5\,\mathrm{PgC\,year^{-1}}$ for forests, grasslands and shrubs, respectively, in North America and Greenland (Figures 5a, b, d and g, respectively).

## 3.4 Sensitivity to spatial resolution

When simulating GPP at global and regional scales, there was little impact from varying spatial resolution ($0.5° \times 0.5°$, $1° \times 1°$ and $2° \times 2°$) (Figure 6). When simulations of GPP were performed at lower spatial resolutions (JULES-WFDEI-GPCC-1degree and JULES-WFDEI-GPCC-2degree; Table 1), the average annual global GPP at $0.5° \times 0.5°$, $1° \times 1°$ and $2° \times 2°$ spatial reso­lutions was $140\,\mathrm{PgC\,year^{-1}}$, $141\,\mathrm{PgC\,year^{-1}}$ and $142\,\mathrm{PgC\,year^{-1}}$, respectively. The percentage differences between JULES and the observation-based GPP estimates (MODIS and FLUXNET-MTE) at the various spatial resolutions are approximately equal with JULES differing from MODIS and FLUXNET-MTE by 8 % and 25 %, respectively, at $0.5° \times 0.5°$ spatial resolution, by 8 % and 26 %, respectively, at $1° \times 1°$ resolution and by 9 % and 26 %, respectively, at $2° \times 2°$ resolution.

The zonal mean of modelled total annual GPP at various spatial resolutions are approximately equal (Figure 6). This in­sensitivity to spatial resolution is also found at regional scales (Figure 5). This insensitivity to spatial resolution is a useful result since it means that model simulations can be performed at $2° \times 2°$ resolution with little difference to model output from the simulations at $0.5° \times 0.5°$ and due to the lower computational cost, model run times (at $2° \times 2°$ resolution) are short (approximately $16\times$ faster than the $0.5° \times 0.5°$ resolution simulations).

## 3.5 Sensitivity to meteorological data set

When JULES was driven with the PRINCETON dataset, simulated global GPP was found to be higher than that simulated using WFDEI-GPCC by $3\,\mathrm{PgC\,year^{-1}}$ on average with the largest differences occurring in the tropics (Figures 6, 7a and 7d). When driven with the PRINCETON dataset (JULES-PRINCETON; Table 1), JULES simulates global GPP with an annual average of $144\,\mathrm{PgC\,year^{-1}}$ for the 2001–2010 period (Figure 7d).

As observed when driving JULES with the WFDEI-GPCC dataset (Figure 2), JULES-PRINCETON simulates higher global GPP than MODIS, FLUXNET-MTE and CARDAMOM at $1° \times 1°$ spatial resolution by 11–29 %. This compares quite well to global GPP simulated by JULES when driven with the WFDEI-GPCC dataset, which had an annual average global GPP of $140\,\mathrm{PgC\,year^{-1}}$. GPP simulated by JULES-WFDEI-GPCC was only higher than that of MODIS, FLUXNET-MTE (both at $0.5° \times 0.5°$ spatial resolution) and CARDAMOM (at $1° \times 1°$ resolution) by 8–25 %. The pattern in zonal mean of total annual GPP simulated by the model (when driven with PRINCETON) is similar to that when driven with WFDEI-GPCC (at $1° \times 1°$ spatial resolution) with differences between JULES-PRINCETON and JULES-WFDEI-GPCC-1degree and the observation-based estimates (MODIS and FLUXNET-MTE) being mostly in the tropics (Figure 6).

There is little difference in simulated GPP when using either WFDEI-GPCC or WFDEI-CRU (which differ only in the precipitation product used) to drive JULES (Figure 4; Figure G.2 in Slevin (2016)). When driven with the WFDEI-CRU dataset, JULES simulates global GPP with an annual average of $142\,\mathrm{PgC\,year^{-1}}$ (the combined GPP of all terrestrial ecosystems) over 2001–2010 (Figure G.3 in Slevin (2016)). This is $2\,\mathrm{PgC\,year^{-1}}$ higher than that simulated when JULES is driven with WFDEI-GPCC ($140\,\mathrm{PgC\,year^{-1}}$). The small differences in global GPP can also found at regional scales in both the tropical and extratropical regions (Figures 4b and c, respectively).

## 4    Discussion

### 4.1    *How do estimates of total annual GPP compare to those from observational datasets? Can JULES capture the seasonal and interannual variability of GPP at global scales?*

At global scales, JULES estimates the annual average GPP (combined GPP of all terrestrial ecosystems) to be $140\,\mathrm{PgC\,year^{-1}}$,
which is greater than MODIS, FLUXNET-MTE and CARDAMOM GPP by 8–25 % (Figure 2). The annual average MODIS,
FLUXNET-MTE and CARDAMOM GPP estimates over 2001–2010 are 112, 130 and 114 $\mathrm{PgC\,year^{-1}}$, respectively (Figure 2). Differences in these estimates are due to differences in forest and grassland GPP in the tropics (Figure 4b). MODIS and
CARDAMOM GPP estimates are similar at global and regional scales since both use MODIS LAI data to determine GPP (Figure 2). In the extratropics, JULES was able to simulate GPP well compared to MODIS, FLUXNET-MTE and CARDAMOM
since its phenology model and associated model parameters may have been designed for temperate regions. When JULES was
driven with the WFDEI-GPCC dataset (at $0.5° \times 0.5°$ spatial resolution), the model was able to capture interannual variability
at the global scale (Figure 3b).

The main difference between JULES and CARDAMOM GPP estimates was found in the tropics with CARDAMOM GPP
being between the two observation-based datasets (Figure 2e). Photosynthesis is also modelled differently in JULES and
CARDAMOM. In JULES, leaf-level photosynthesis is calculated as the minimum of three limiting rates which is then scaled
up to canopy level using the sum of the leaf-level fluxes in each canopy layer, scaled by leaf area (Clark et al., 2011). In
CARDAMOM, GPP is calculated as a function of LAI, air temperature and radiation using the Aggregated Canopy Model
(Williams et al., 1997, ACM). ACM is an emulator of the Soil Plant Atmosphere (SPA) model and uses a set of equations to
simulate daily GPP estimates produced by SPA (Williams et al., 1996).

JULES simulates lower GPP than MODIS, FLUXNET-MTE and CARDAMOM at 15°N–30°N (Figures 6 and 8). This
large negative bias in JULES was due to the incorrect simulation of GPP in subtropical regions such as Mexico and southern
China (Figure 8a, b and c). The total annual MODIS and FLUXNET-MTE GPP estimates for 2001–2010 are higher than
that simulated by JULES by 1% and 7%, respectively, for Mexico, with CARDAMOM GPP estimates for the same period
being lower than JULES GPP by 6%. The tropical GPP fluxes for forests, grasslands and shrubs were further subdivided into
two regions; (1) the tropics at 30°S–15°N (Figure 4d) and (2) Mexico (Figure 4e). The total (summed over 10 years) JULES
GPP was similar for the two tropical regions at 30°S–30°N and 30°S–15°N with positive biases in forest and grassland GPP
(Figures 4b and d). GPP in Mexico was similar for forests and grasslands with differences in shrub GPP (Figure 4e). The
negative bias in JULES GPP in the subtropics is due to low LAI simulated by the model compared to MODIS (Figure 5.10
in Slevin (2016); Figure 8d). MODIS LAI is used as input when generating the MODIS, FLUXNET-MTE and CARDAMOM
GPP estimates.

One of the major vegetation types in the subtropics is drought-deciduous plants (drought-deciduous plants lose their leaves
during the dry season or periods of dryness as opposed to temperate deciduous plants which lose their leaves during periods
of cold weather) and JULES does not contain this PFT. Drought-deciduous plants can be found in the seasonally dry tropical
forests of Mexico, Central America, northwestern South America and southern China. The implementation of a drought-

deciduous shrub PFT would help improve simulated GPP in these regions. In JULES, phenology is updated once per day by multiplying the annual maximum LAI by a scaling factor, which is calculated using temperature-dependent leaf turnover rates. Leaf turnover rates are a function of surface air temperature and increase when the temperature drops below a certain value (this varies depending on the PFT). While this is suitable for deciduous broadleaf forests in temperate regions, such as Northern Europe, it will lead to inaccurate modelled LAI for drought-deciduous forests. Instead of modifying modelled LAI using a temperature-derived scaling factor, the scaling factor could be calculated by using periods of dryness as the controlling factor.

In general, when JULES was driven with the WFDEI-GPCC dataset at global scales (JULES-WFDEI-GPCC-1degree), it was found that simulated photosynthesis was Rubisco-limited (Figures 5.6 and 5.7 in Slevin (2016)). Under saturated irradiance and limited atmospheric $CO_2$ concentrations, the Rubisco limiting rate is the main limiting factor (Marcus et al., 2008). Since the multi-layer approach for radiation interception and scaling from leaf-level to canopy-level photosynthesis was used by JULES in this study, the model simulates competition between Rubisco-limited and light-limited photosynthesis for each canopy layer (Clark et al., 2011). This means that lower in the canopy, there is increased light limitation and in the upper layers of the canopy, Rubisco limitation dominates (Clark et al., 2011). Overall, the percentage of model gridboxes that were found to be Rubisco-limited was high (40– 100%), whereas the percentage of model gridboxes that were found to be light-limited were small (0–20%) (Figures 5.7 and 5.8, respectively, in Slevin (2016)). A description of the methods used to determine which limiting rate dominates each model gridbox when calculating potential leaf-level photosynthesis is provided in Appendix F of Slevin (2016).

In regions dominated by grasses and shrubs, photosynthesis was found to be transport-limited (Figure 5.6 in Slevin (2016)), which refers to the rate of transport of photosynthetic products (for C3 plants) and PEPCarboxylase limitation (for C4 plants). Transport limitation occurs mostly in Northern Eurasia and North America during the Spring and Summer months (March– September) and during the Autumn and Winter months (October–February) in Central Asia (Figures 5.6 and 5.9 in Slevin (2016)).

*4.2   How do fluxes of GPP simulated by JULES compare for various biomes at the global and regional scales?*

At global scales, JULES (JULES-WFDEI-GPCC) simulated average annual GPP to be 68, 62 and $9 \, \mathrm{PgC \, year^{-1}}$ for forests, grasslands and shrubs, respectively. Simulated GPP for forests is higher than that that calculated by Beer et al. (2010) (sum of the values for tropical, temperate and boreal forests) with average annual GPP being $59 \, \mathrm{PgC \, year^{-1}}$. Since Beer et al. (2010) provides average annual GPP values for tropical savannahs and grasslands, temperate grasslands and shrublands and croplands, these are summed in order to obtain average annual global GPP for grasslands and shrubs $54.6 \, \mathrm{PgC \, year^{-1}}$, which is lower than the JULES grasslands and shrubs simulated total value of $71 \, \mathrm{PgC \, year^{-1}}$.

Differences between MODIS and CARDAMOM estimates of average annual GPP are similar with MODIS simulating average annual GPP to be 52.3, 50.1 and $9.4 \, \mathrm{PgC \, year^{-1}}$ for forests, grasslands and shrubs, respectively, and CARDAMOM simulating average annual GPP to be 56.5, 48.6 and $9.2 \, \mathrm{PgC \, year^{-1}}$ for forests, grasslands and shrubs, respectively (Figure 4a). The MODIS and CARDAMOM GPP estimates are similar due to MODIS LAI being assimilated into CARDAMOM GPP

simulations. FLUXNET-MTE GPP is higher than the MODIS and CARDAMOM estimates for all biomes (Figure 4). JULES simulates higher GPP than MODIS, FLUXNET-MTE and CARDAMOM at global scales and this was found to be due to higher GPP simulated by JULES for forests and grasslands in the tropics (Figure 4b). The average annual GPP for shrubs was similar between model (JULES and CARDAMOM) and observation-based (MODIS and FLUXNET-MTE) estimates for all three regions (Figure 4).

A simple sensitivity study of the model to changes in climate (surface (2m) air temperature, precipitation and atmospheric $CO_2$ concentrations) when simulating GPP at global and regional scales for 2000–2010 was performed(Slevin, 2016). Only changes to one climate variable were made at a time due to complex interactions associated with multiple changes in climatic factors resulting in complex non-linear ecosystem responses which can be difficult to explain. JULES GPP was found to be sensitive to changes in all three climate variables with modelled LAI only sensitive to changes in surface air temperature (Slevin, 2016). At the regional scale, for model simulations with varying air temperature, GPP increased with increasing temperature in the extratropics, but decreased with increasing temperature in the tropics. Model simulations with varying precipitation at regional scales show the same trend as those at global scales with GPP increasing with increasing precipitation and decreasing with decreasing precipitation except for the magnitude of the effect observed. More detailed information on the sensitivity study is provided in Chapter 6 of the PhD thesis of Darren Slevin(Slevin, 2016).

There are two possible reasons for the larger simulated GPP in the tropics at 30°S–15°N. Firstly, the high bias in the tropics during December, January, February and March (Figures 2e and 3b) imply that JULES GPP is overestimated in the tropical wet season. The lower air temperatures and higher soil moisture availability during the wet season as a result of the meteorological data leads to higher simulated GPP in these regions. Secondly, the higher tropical GPP is due to the incorrect simulation of GPP by the PFTs in the version of JULES (version 3.4.1) used in this study. In this version, the PFT used to represent tropical forests is the broadleaf tree, which is used to simulate GPP in both tropical and temperate regions. This means that the model parameters used for the broadleaf tree PFT may not be suitable for simulating GPP in the tropics. The addition of extra PFTs more suited to tropical regions, such as tropical broadleaf evergreen (in version 4.2) and a drought-deciduous PFT, and a phenology model which can simulate LAI in tropical regions would both improve GPP simulations.

By dividing the global land area into seven regions (Table 2), it was found that for all three tropical regions (Central and South America, Africa and South and South-East Asia), JULES overestimated total annual GPP for forests, grasslands and shrubs (Figures 5c, e, and f). In the four extratropical regions (Europe, Northern Asia, Extratropical Southern Hemisphere and North America and Greenland), JULES simulated similar GPP to MODIS, FLUXNET-MTE and CARDAMOM for the three biomes in Europe and the Extratropical Southern Hemisphere (Figures 5a and d), with the exception of Northern Asia, and North America and Greenland, where the model is either equal to or lower than all three datasets (Figures 5b and g).

One possibility for the higher GPP estimates in Northern Asia and North America and Greenland is due to a different land cover map being used by the observation-based datasets (MODIS and FLUXNET-MTE) and JULES. The PFT fractions specified in the land cover map used by JULES were used to calculate the GPP estimates for the forests, grasslands and shrubs biomes for MODIS, FLUXNET-MTE and CARDAMOM. So for a particular gridbox, the land cover map of JULES may specify a shrub, but in the land cover map used by MODIS or FLUXNET-MTE, it may be a needleleaf tree. Another possibility

is that shrubs in these northern regions have adapted to the cold environment and the lower surface air temperatures have a lesser impact on photosynthesis than they do on shrubs in warmer climates. The addition of a shrub PFT to JULES which is adapted to cold climates may improve GPP estimates in these regions.

### 4.3 How sensitive are fluxes of GPP to the spatial resolution of the model?

5  JULES was insensitive to spatial resolution with average annual global GPP being $140\,\mathrm{PgC\,year^{-1}}$, $141\,\mathrm{PgC\,year^{-1}}$ and $142\,\mathrm{PgC\,year^{-1}}$ at $0.5° \times 0.5°$, $1° \times 1°$ and $2° \times 2°$ spatial resolutions, respectively. This pattern was also observed in the zonal mean of total annual GPP (Figure 6). The insensitivity of the model to spatial resolution at the global scale was also observed at the regional scale when comparing simulated GPP fluxes for forests, grasslands and shrubs in the tropics and extratropics (Figure 5).

10  Little research has been performed on the effects of spatial resolution on JULES simulations (as well as other LSMs). Studies using atmospheric chemistry models have shown that the spatial resolution of the input meteorological data can affect model output (Ito et al., 2009; Pugh et al., 2013; Schaap et al., 2015). The results found here agree with those from Compton and Best (2011). Compton and Best (2011) showed that JULES was insensitive to spatial resolution when the WFD dataset was regridded from half-degree to 1-degree and 2-degree when simulating the terrestrial hydrological cycle. It was found that 15 spatial resolution had little or no effect on simulations of global mean total evaporation and total runoff. However, the study showed that JULES was sensitive to temporal resolution when simulating the same hydrological components.

Using a different soil ancillary dataset or land cover map (which specifies the PFT fractions) may have a larger impact than changing the spatial resolution. The regridding method used in this study was the conservative method, which preserves the same information when interpolating from $0.5° \times 0.5°$ to $1° \times 1°$ and $2° \times 2°$ spatial resolutions, and results in only small 20 differences in global GPP between the model simulations with varying spatial resolution. These small differences are due to differences in the PFT fractions of the land cover map after regridding.

### 4.4 Is the meteorological dataset used to drive the model important at the global scale?

When JULES was driven with the PRINCETON dataset at $1° \times 1°$ spatial resolution (Table 2), the annual average global GPP was slightly higher by $3\,\mathrm{PgC\,year^{-1}}$ than that simulated by JULES when driven with the WFDEI-GPCC dataset at 25 the same resolution. In general, differences in GPP fluxes for model simulations driven using WFDEI-GPCC and PRINCE-TON are mainly in the deep tropics (at 5°N–5°S) with JULES-WFDEI-GPCC-1degree simulating higher GPP than JULES-PRINCETON and in the extratropics at 30°N–60°N, JULES-PRINCETON simulates slightly higher GPP (Figures 6 and 5).

The higher simulated GPP at 5°N–5°S in the Amazonian, African and South-East Asian tropics in the WFDEI-GPCC driven simulation is due to lower surface air temperatures (Figures G.6a and c in Slevin (2016)) and higher precipitation (Figures 30 G.6b and d in Slevin (2016)) in these regions(Slevin, 2016). In extratropical regions, such as northern Eurasia (above 60°N), there are differences in the meteorological datasets with slightly higher downward shortwave radiation fluxes and surface air temperatures in the PRINCETON dataset with little difference between the JULES simulations driven with either WFDEI-GPCC or PRINCETON in this region (Figure 6).

Other studies have shown that the meteorological dataset used to drive LSMs is a large source of uncertainty in global land surface modelling (Hicke, 2005; Jung et al., 2007; Poulter et al., 2011). Different methods are used to create time series of global gridded climate data in order to drive LSMs and this can introduce uncertainty that can propagate through model simulations (Zhao et al., 2006). Even at the point scale, differences in simulated GPP were observed when driving JULES with the WFDEI-GPCC and PRINCETON datasets (Slevin et al., 2015). As in this study, it also occurred in the tropics.

When JULES was driven with the PRINCETON dataset, simulated photosynthesis was mostly Rubisco-limited (Figures 5.25 and 5.6 in Slevin (2016)). This is a similar result to when JULES was driven with the WFDEI-GPCC dataset. When model gridbox fractions were compared between the WFDEI-GPCC and PRINCETON driven model simulations, photosynthesis in the PRINCETON driven simulation was more Rubisco-limited than that when driven with WFDEI-GPCC (Figure 5.26 in Slevin (2016)) and is due to higher surface air temperatures (used in calculating photosynthesis for the Rubisco limiting rate) in the PRINCETON dataset (Figures G.6a and c in Slevin (2016)).

Since the WFDEI-GPCC dataset has lower downward shortwave radiation than PRINCETON, photosynthesis in the WFDEI-GPCC driven simulation was more light-limited than the PRINCETON simulation (Figures 5.27, G.5a and G.5c in Slevin (2016)). The number of model gridboxes in which transport limitation dominated in the PRINCETON driven simulation was less than that for the JULES-WFDEI-GPCC-1degree model simulation (Figure 5.28 in Slevin (2016)). There is a pronounced geographical variation with the WFDEI-GPCC driven simulation being more transport-limited in the tropics and the PRINCETON driven simulation being more transport-limited in the extratropics (Figure 5.28 in Slevin (2016)). This is likely due to the lower surface air temperatures in the WFDEI-GPCC which results in lower potential leaf-level photosynthesis for C3 and C4 plants in the extratropics and tropics, respectively.

In this study, the model simulations were performed with prescribed PFTs (i.e. no vegetation competition). If competition between PFTs was allowed (i.e. vegetation competition), the annual average global GPP would be higher by 15 % and 17 %, for the WFDEI-GPCC and PRINCETON driven simulations, respectively (Figures 7b and e). Higher GPP occurred mostly in Europe, southeastern US, and in the tropical regions of Central and South America, Africa and South and South-East Asia (Figures 7c and f). This increased GPP in tropical regions is due to the tree-shrub-grass dominance heirachy in TRIFFID with dominant types (trees) limiting the expansion of subdominant types (shrubs and grasses). In savanna regions, such as the Sudanian Savanna, which stretches from the Atlantic Ocean in the west to the Ethiopian Highlands in the east of Africa, and northern Australia, there is higher GPP with prescribed PFTs (Figures 7c and f). These are also fire-prone regions. The version of JULES used in this study has no fire module or deforestation implemented and TRIFFID may overestimate woody cover and therefore GPP.

In terms of global GPP, the WFDEI-GPCC and PRINCETON driven simulations produce similar increases (Figures 7b and e). However, the spatial pattern is slightly different with higher GPP simulated in the Amazon region when JULES was driven with the WFDEI-GPCC dataset and higher GPP in southern Brazil and Argentina and Southeast Asia when JULES was driven with the PRINCETON dataset (Figures 7c and f). The spatial pattern of simulated GPP is more sensitive to the meteorological data than the annual average global GPP if competition between PFTs is allowed. This may be due to compensating differences in the sensitivity of the model to the two meteorological datasets.

## 5 Conclusions

An evaluation of JULES was performed at global and regional scales with simulated GPP compared to global gridded ($0.5° \times 0.5°$ spatial and monthly temporal resolution) estimates of GPP derived from upscaled FLUXNET observations (FLUXNET-MTE), satellite observations from the MODIS sensor and that produced by the CARDAMOM data assimilation framework. JULES simulated higher average annual global GPP than FLUXNET-MTE, MODIS and CARDAMOM but at regional scales, differences arose in the tropics. It was found that JULES was able to capture interannual variability at the global scale.

Differences in GPP between JULES and the benchmarking datasets (FLUXNET-MTE, MODIS and CARDAMOM) at 15°N–30°N is due to higher FLUXNET-MTE, MODIS and CARDAMOM GPP in regions such as Mexico and southern China. These differences may be due to a lack of drought-deciduous PFTs in JULES. The inclusion of these PFTs could improve GPP simulations at latitude 15°N-30°N. By dividing the global land area into seven regions, it was found that all three tropical regions (Central and South America, Africa and South and South-East Asia) contribute to model-observation differences at the global scale compared to FLUXNET-MTE and MODIS. The model can reasonably reproduce GPP estimates in the four extratropical regions (Europe, Northern Asia, North America and Greenland, and the extratropical Southern Hemisphere).

Improved GPP simulations in the tropics could be attained with the introduction of more PFT classes and their associated model parameters. In the version of JULES used in this study (3.4.1), each model grid box is composed of nine different surface types and five of these are PFTs. Since model version 4.2, each JULES gridbox contains nine PFTS (tropical broadleaf evergreen, temperate broadleaf evergreen, broadleaf deciduous, needleleaf evergreen, needleleaf deciduous, C3, C4, evergreen shrub, deciduous shrub) (Harper et al., 2016). In addition to these PFTs, a phenology model which can simulate LAI in both temperate and tropical regions, would help to reduce differences between model simulated and observation-based estimates of GPP in the dry and wet tropics.

When JULES was driven at the global and regional scale with the WFDEI-GPCC dataset at various spatial resolutions ($0.5° \times 0.5°$, $1° \times 1°$ and $2° \times 2°$), it was found that the model was insensitive to spatial resolution. Similar results were shown by Compton and Best (2011) when simulating components of the terrestrial hydrological cycle. Differences between high ($0.5° \times 0.5°$) and low ($2° \times 2°$) spatial resolution simulations of GPP are very similar. This means that low spatial resolution model simulations at these scales can be performed in place of high resolution when simulating GPP and results in shorter model run times.

The meteorological dataset used to drive LSMs at the global scale is an important source of model uncertainty (Poulter et al., 2011). By using a different meteorological dataset (PRINCETON) to drive the model, it was found that simulated GPP was similar to that when the model was driven with the WFDEI-GPCC dataset (at $1° \times 1°$ spatial resolution) with exceptions to this being in the tropics and the northern extratropics. Differences in the tropics are due to lower surface air temperatures and higher precipitation (and therefore increased soil moisture availability) in the WFDEI-GPCC dataset and in the extratropics, due to higher air temperatures in the PRINCETON dataset. Photosynthesis in the WFDEI-GPCC and PRINCETON driven simulations were Rubisco-limited. The model simulations in this study were largely performed with prescribed PFTs (i.e. no competition between PFTs was allowed). With competition between PFTs, the annual average global GPP was higher by 15 %

and 17 %, for the WFDEI-GPCC and PRINCETON driven simulations, respectively, with the spatial pattern of simulated GPP more sensitive to the meteorological data used.

The three benchmarking datasets all contain sources of error. Since observations of GPP do not exist at global scales, the MODIS and FLUXNET-MTE datasets are referred to as observation-based estimates of GPP as they are generated using observations and models. CARDAMOM may contain significant error from the assimilated data and model structure (number of pools, fire resilience of ecosystems), but so do the empirically based FLUXNET-MTE data (up-scaling of a partitioning algorithm) and MODIS GPP (a model based on PFT specific light-use efficiency). The advantage of CARDAMOM is that it is a process-based model and it ensures that the whole ecosystem functioning is coherent, while the observation-based datasets are only empirically based representations of GPP. In Figure S4 of the Supplementary Information of Bloom et al. (2016), there is a detailed study of the sensitivity of CARDAMOM to these various factors at 4 selected pixels representing temperate, boreal, wet and dry tropical ecosystems. Overall, there is not much difference in retrieved parameters because of the large error/uncertainty terms used when computing the likelihood.

In general, differences between JULES GPP and the benchmarking datasets (FLUXNET-MTE, MODIS and CARDAMOM) occur mostly in the tropics with differences at 15°N–30°N possibly due to a lack of drought-deciduous PFTs in JULES. When JULES was driven with different meteorological datasets (WFDEI-GPCC and PRINCETON), the WFDEI-GPCC driven model simulations estimated higher GPP in the tropics (at 5°N–5°S) and the PRINCETON driven model simulations higher GPP in the extratropics (at 30°N–60°N). The meteorological dataset used to drive JULES was found to be a source of model uncertainty in the tropics, though this may be due to model error. By using a different precipitation product (WFDEI-CRU), differences in JULES GPP were very small. Finally, when model simulations of GPP were performed at various spatial resolutions (0.5° × 0.5°, 1° × 1° and 2° × 2°), JULES was found to be insensitive to spatial resolution.

## 6   Code and/or data availability

The JULES model code (v3.4.1) is stored at the Met Office Science Repository Service in the JULES repository (https:// code.metoffice.gov.uk/trac/jules) and access to the code can be requested from the official website of JULES (https://jules. jchmr.org/software-and-documentation). The outputs from the JULES model simulations reported in this paper have been deposited online at DataShare, the University of Edinburgh's digital repository of multidisciplinary research datasets, http: //dx.doi.org/10.7488/ds/1461. The ancillary data (soil, vegetation, etc.) used for these simulations have also been deposited on DataShare (http://dx.doi.org/10.7488/ds/1995).

*Author contributions.* D.S., S.F.B.T. and M.W. designed the research. D.S., J.-F.E. and A.A.B performed the model simulations. D.S., S.F.B.T. and M.W. analysed the data. D.S. prepared the manuscript with contributions from all co-authors.

*Acknowledgements.* The authors would like to thank the anonymous reviewers and editor for their valuable comments which improved the quality of the paper. We would like to gratefully acknowledge the School of GeoSciences, University of Edinburgh, which provided D.

Slevin with a School Scholarship that allowed him to pursue his PhD research, during which this work was carried out. Part of this work was conducted at the Jet Propulsion Laboratory, California Institute of Technology under a contract with NASA and supported by the Natural Environment Research Council (NERC) and the UK National Centre for Earth Observation (NCEO). This work would not have been possible without help from the following people. Rich Ellis at the Centre for Ecology and Hydrology (CEH) provided the soil and vegetation ancillary files and advice on modifying the namelist files in order to set up the model simulations to run on a 2d grid. Doug Clark (also at CEH) provided advice on model spin-up at global scales. Information on the snow schemes used by JULES and advice on which one to use at global scales was provided by Richard Essery. Graham Weedon provided invaluable advice on using the WFDEI meteorological dataset.

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

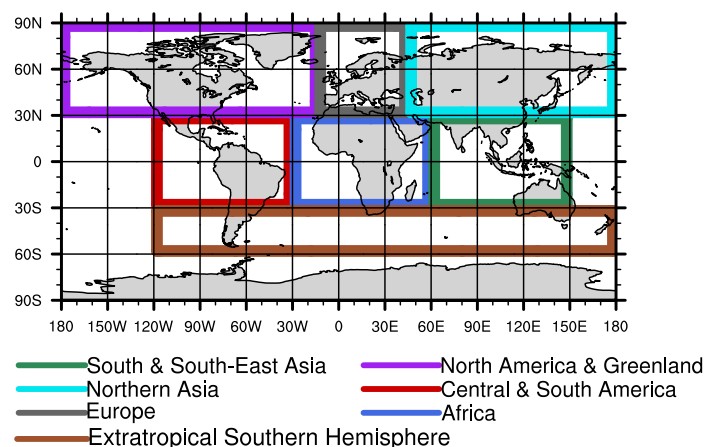

**Figure 1.** Map showing the regions specified in Table 2.

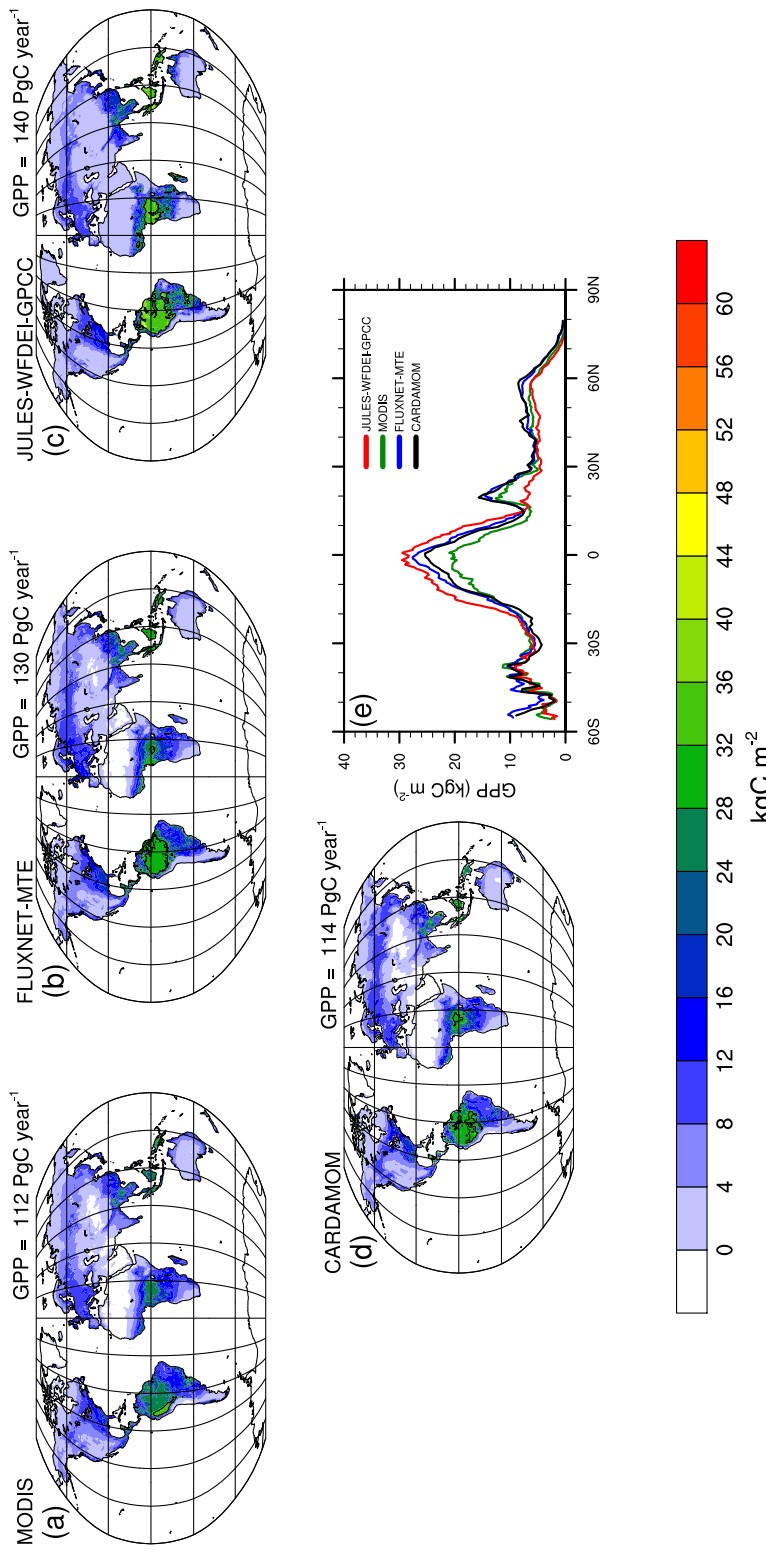

**Figure 2.** Total annual and zonal mean model simulated (JULES-WFDEI-GPCC), observed (FLUXNET-MTE and MODIS) and CARDAMOM GPP fluxes for the 2001–2010 period at the global scale. JULES, FLUXNET-MTE and MODIS GPP are at 0.5° × 0.5° spatial resolution and CARDAMOM is at 1° × 1° resolution. (**a**), (**b**), (**c**) and (**d**) show the total annual GPP of JULES-WFDEI-GPCC, FLUXNET-MTE, MODIS and CARDAMOM GPP, respectively. At the top right of each map subplot, the average global annual GPP for 2001–2010 is displayed. (**e**) shows the zonal mean of the total annual JULES-WFDEI-GPCC, FLUXNET-MTE, MODIS and CARDAMOM GPP, respectively. Included in each map subplot are contour lines for the tropical regions.

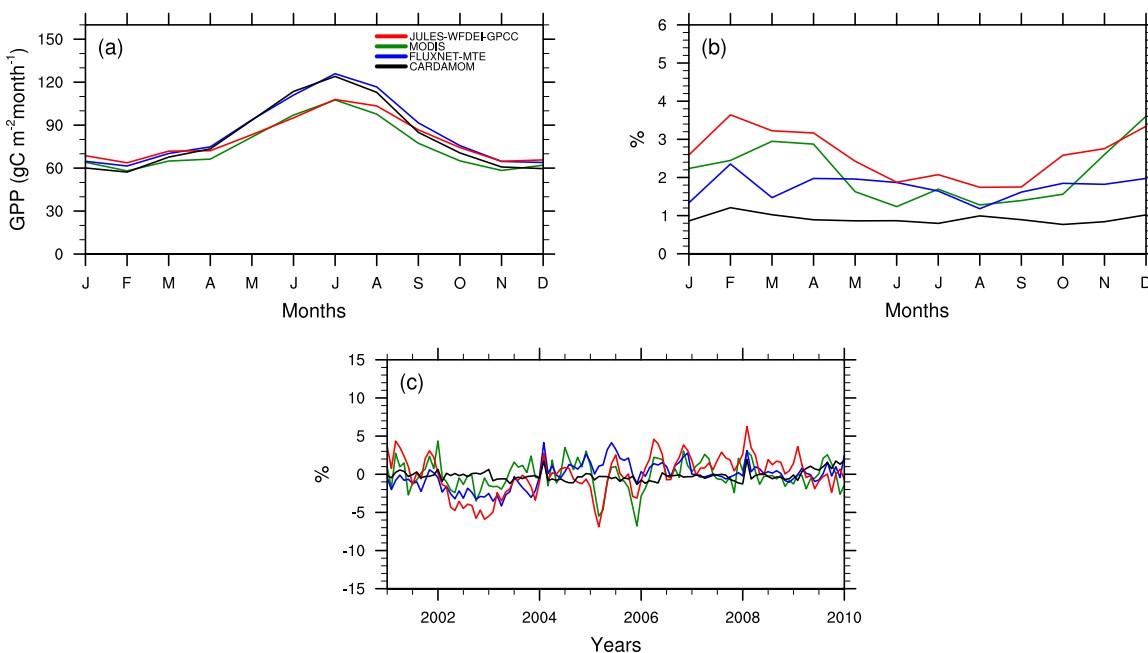

**Figure 3.** Comparison of JULES, observation-based (FLUXNET-MTE and MODIS) and CARDAMOM (Table 1) GPP fluxes for the 2001–2010 period at global scales. **(a)** shows the global average of the mean monthly GPP, **(b)** shows the coefficient of variation (CV) expressed as percentages of the mean monthly GPP and **(c)** shows the monthly anomalies expressed as percentages of the mean monthly GPP for each month.

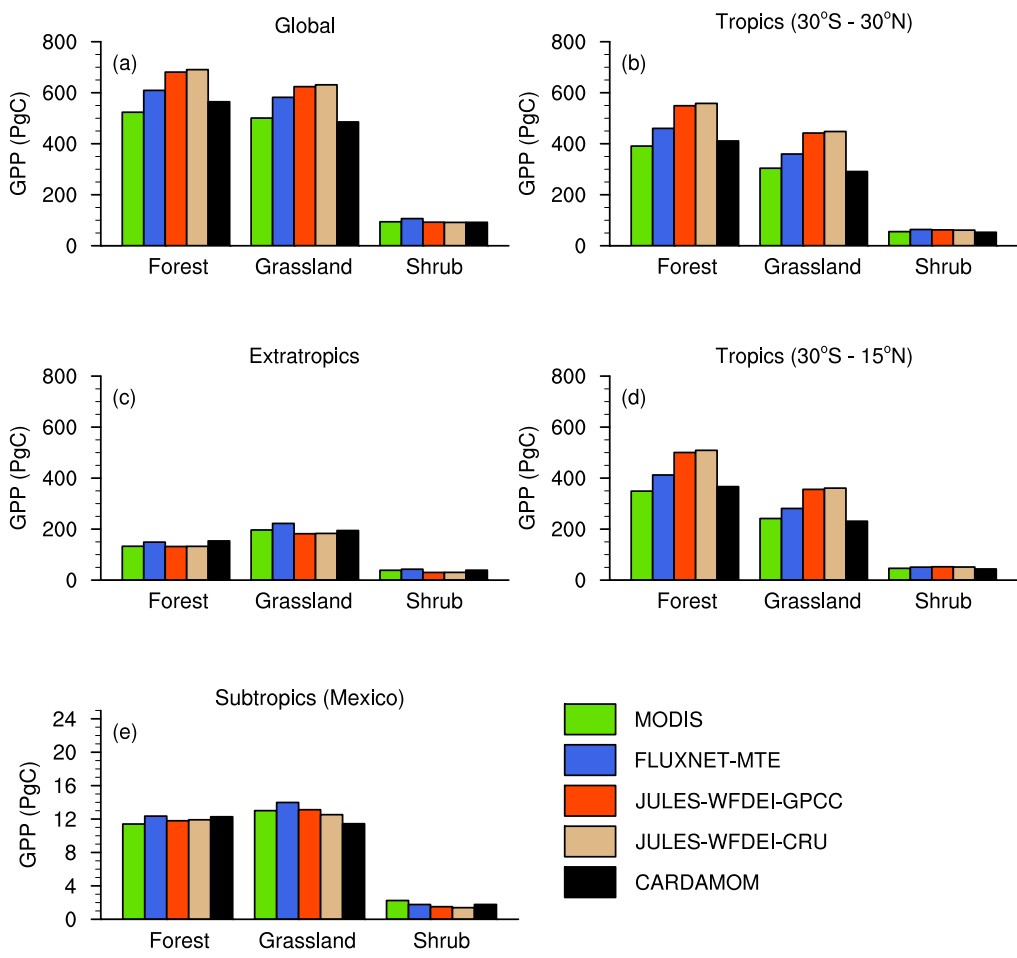

**Figure 4.** Total (summed over 10 years) model simulated (JULES-WFDEI-GPCC, JULES-WFDEI-CRU and CARDAMOM), observation-based (FLUXNET-MTE and MODIS) GPP fluxes for the 2001–2010 period at global and regional scales (tropics, subtropics and extratropics) for 3 biome types (Forest, Grassland and Shrub). **(a)** shows the global total annual GPP, **(b)** for the tropics ($30°$S–$30°$N), **(c)** the extratropics ($30°$N–$90°$N and $30°$S–$90°$S), **(d)** the tropics at $30°$S–$15°$N and **(e)** the subtropics at $15°$N–$30°$N for forests, grasslands and shrubs.

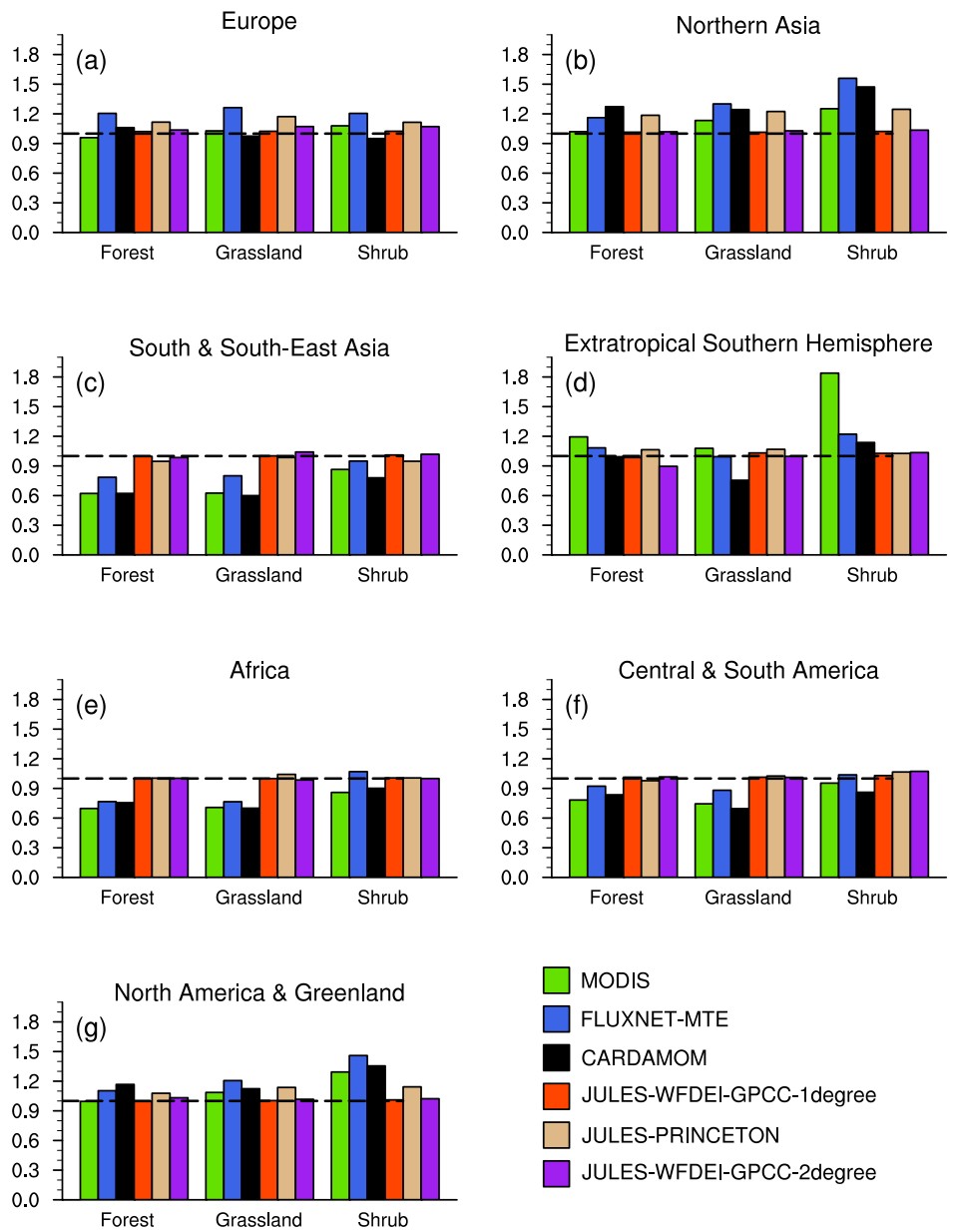

**Figure 5.** Total annual model simulated (JULES-WFDEI-GPCC-1degree, JULES-PRINCETON, CARDAMOM and JULES-WFDEI-GPCC-2degree) and observed (FLUXNET-MTE and MODIS) GPP fluxes for the 2001–2010 period normalised by model simulated (JULES-WFDEI-GPCC) GPP for various regions (Table 2) for 3 biome types (Forest, Grassland and Shrub). **(a)** shows normalised GPP for Europe, **(b)** for Northern Asia, **(c)** for South & South-Asia, **(d)** for extratropical Southern Hemisphere, **(e)** for Africa, **(f)** for Central & South America and **(g)** for North America & Greenland. The dotted line at y=1 represents where the model and observations match.

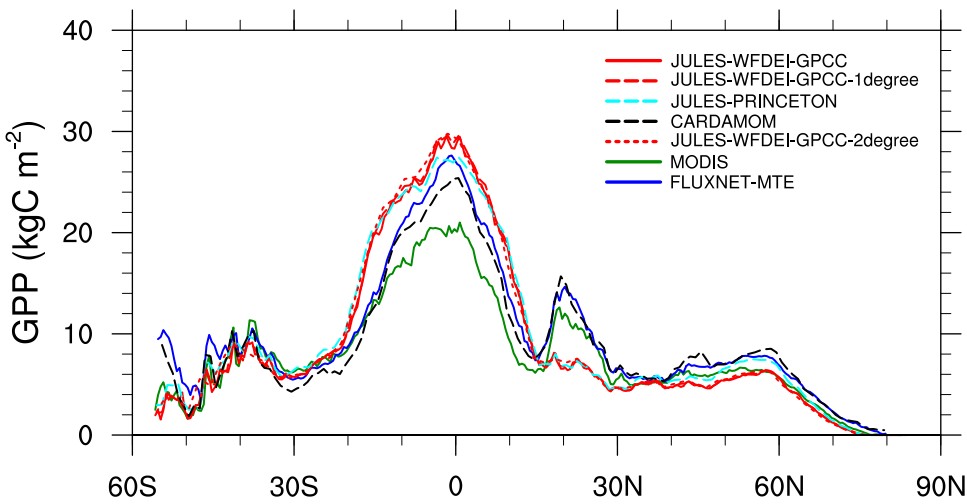

**Figure 6.** Zonal mean of total annual model simulated (JULES-WFDEI-GPCC, JULES-WFDEI-GPCC-1degree, JULES-PRINCETON, CARDAMOM and JULES-WFDEI-GPCC-2degree) and observed (FLUXNET-MTE and MODIS) GPP fluxes for 2001–2010. JULES-WFDEI-GPCC, FLUXNET-MTE and MODIS are at $0.5° \times 0.5°$ spatial resolution.

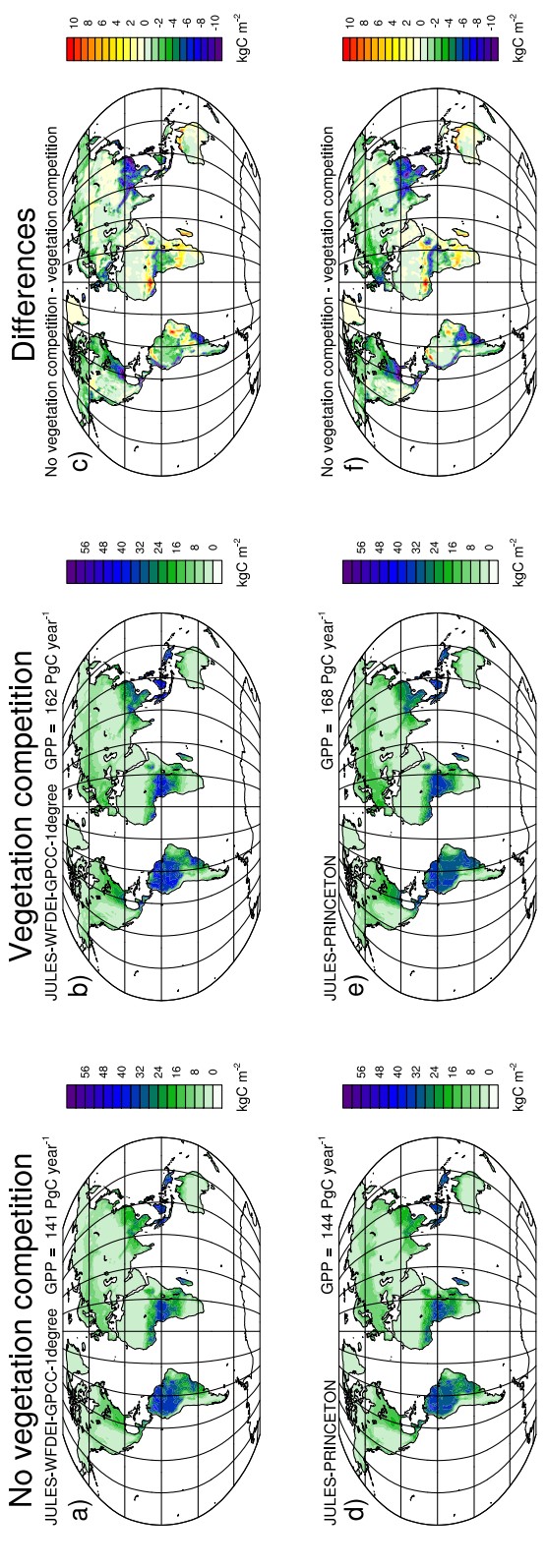

**Figure 7.** Total annual model simulated (JULES-WFDEI-GPCC-1degree and JULES-PRINCETON) GPP when simulations were performed with prescribed PFTs (vegetation competition switched off) and with different PFTs competing against each other (vegetation competition switched on) for 2001−2010. (**a**) and (**b**) show the total annual JULES-WFDEI-GPCC-1degree GPP with vegetation competition switched off and on, respectively, and (**c**) shows the difference. (**d**) and (**e**) show the total annual JULES-PRINCETON GPP with vegetation competition switched off and on, respectively, and (**f**) shows the difference. At the top right of (**a**), (**b**), (**d**) and (**e**), the average annual global GPP for 2001−2010 is displayed.

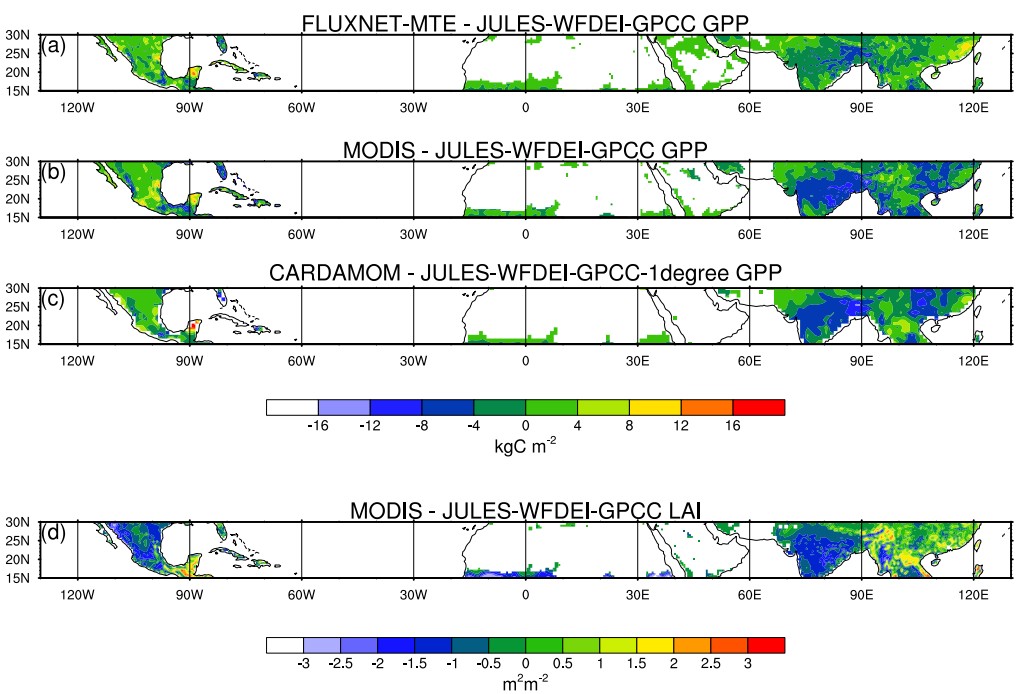

**Figure 8.** Difference in total annual GPP between JULES-WFDEI-GPCC and the observation-based (FLUXNET-MTE and MODIS) and CARDAMOM estimates of GPP and in monthly mean LAI between JULES-WFDEI-GPCC and MODIS at latitudes $15°$N-$30°$N for the 2001–2010 period. **(a)** shows the difference in GPP between FLUXNET-MTE and JULES, **(b)** between MODIS and JULES and **(c)** between CARDAMOM and JULES. **(d)** shows the difference in LAI between MODIS and JULES. A positive change in GPP means the observation-based estimates (FLUXNET-MTE and MODIS) or CARDAMOM estimate are higher than the model and in LAI means MODIS LAI is higher than JULES.

**Table 1.** Types of global scale model simulations performed.

| Model simulations | Meteorological forcing | Spatial resolution | Grid dimensions[a] |
|---|---|---|---|
| JULES-WFDEI-GPCC | WFDEI-GPCC | $0.5° \times 0.5°$ | $720 \times 360$ |
| JULES-WFDEI-CRU | WFDEI-CRU | $0.5° \times 0.5°$ | $720 \times 360$ |
| JULES-WFDEI-GPCC-1degree | WFDEI-GPCC | $1° \times 1°$ | $360 \times 180$ |
| JULES-PRINCETON | PRINCETON | $1° \times 1°$ | $360 \times 180$ |
| JULES-WFDEI-GPCC-2degree | WFDEI-GPCC | $2° \times 2°$ | $180 \times 90$ |

[a] Grid dimensions are given as the number of grid boxes in the longitudinal direction by the number of grid boxes in the latitudinal direction.

**Table 2.** List of regions used. Only land grid points are used in the analysis.

| Name | Latitude (°) | Longitude (°) |
|------|--------------|---------------|
| Europe | 30N–90N | 15W–45E |
| Northern Asia | 30N–90N | 45E–180E |
| South & South-East Asia | 30S–30N | 60E–150E |
| Extratropical Southern Hemisphere | 60S–30S | 120W–180E |
| Africa | 30S–30N | 30W–60E |
| Central & Southern America | 30S–30N | 120W–30W |
| North America & Greenland | 30N–90N | 180W–15W |