# Peer review of "Global Evaluation of Gross Primary Productivity in the JULES Land Surface Model v3.4.1"

_Geoscientific Model Development, 2016_

## Short Comment (SC1) · 26 Sep 2016

Dear authors,

In my role as Executive editor of GMD, I would like to bring to your attention our Editorial version 1.1:

http://www.geosci-model-dev.net/8/3487/2015/gmd-8-3487-2015.html

This highlights some requirements of papers published in GMD, which is also available on the GMD website in the 'Manuscript Types' section:

http://www.geoscientific-model-development.net/submission/manuscript_types.html

In particular, please note that for your paper, the following requirements have not been met in the Discussions paper:

- "The main paper must give the model name and version number (or other unique identifier) in the title."

- "If the model development relates to a single model then the model name and the version number must be included in the title of the paper. "

- Inclusion of Code and/or data availability sections is mandatory for all papers and should be located at the end of the article, after the conclusions, and before any appendices or acknowledgments. For more details refer to the code and data policy.

Even though your paper is an "model evaluation paper" and not a "model development paper", it would be good to give the version number of the evaluated JULES model version in the title. Additionally, please add a line about the availability of the JULES model code in the "Code availability" section upon the revised submission of your article to GMD.

Yours,

Astrid Kerkweg
* * *

---

## Short Comment (SC2) · 27 Sep 2016

The version number of the JULES model evaluated in this manuscript has been added to the title. Information on downloading this version of the JULES model has been added to the "Code and/or data availability" section of the manuscript (Page 16, lines 20-25).

Please also note the supplement to this comment:

[revised manuscript text omitted]

---

## Referee Comment (RC1) · Anonymous Referee #1 · 24 Oct 2016

Title: Global Evaluation of Gross Primary Productivity in the JULES Land Surface Model

The authors confirmed the performance of JULES version 3.4.1 in this study. The main analyses are 1) evaluating an effect of different biome type on GPP, 2) comparing GPP among 0.5, 1 and 2-degree grid resolution, 3) examining GPP using three kinds of climate dataset. By using satellite observations, the model estimates were assessed. Unfortunately, I feel like it is a summary of technical reports. Much improvement can still be made to make it clearer and more concise.

General comments; I can't understand the novelty of this manuscript. I agree that the novelty is the performance confirmation of JULES. Please rethink why the authors would like to show others the original results via this manuscript. And, for all things,

if the content is related to just JULES unique performance confirmation, it might not directly help the reader's scientific knowledge. At such times the authors need to improve the explanation by changing the standpoint. Please rewrite the manuscript to serve to help the readers in getting maximum benefit from what the authors revealed.

Please organize all the information of the model introduction. The authors wrote them in 1. Introduction section and 2.1 Model description. Naturally the 2.1 section should be included contents directly related to this study's discussion, and omit the explanation that had little to do with this study. For example, the authors wrote the interminable explanation for the GPP calculation method, but the reader can understand several author statements in discussion section without such knowledge; there is no explanation about spatial resolution as model structure...etc.

Please more explain why the authors used different climate dataset. What of the JULES GPP estimate do the authors reveal? Why did you examine just sensitivity to each dataset? (why didn't you choose the sensitivity to each meteorological parameter?) Please add the comparison among three climate datasets into results. I can't understand the impact of climate dataset on GPP (e.g., fig. 2, 3...), because I don't know the difference of the climate dataset specific feature related to this study. Moreover, please add the explanation of the relationship between JULES and the meteorological parameters in 2.1 Model description section. It means, the reader would like to know the model structural interpretation in discussing what types of calculation approach to choose.

The authors should organize first and second paragraph of "1. Introduction". The authors should integrate the two paragraphs into one. P1 L19-20: delete the sentence (Changes in atmospheric CO2...). P2 L2 and L4: "location of" -> reservoir in? P2 L3: "Changes in the land surface" is not clear. P2 L7: "models and observations (Friedlingstein" -> the existing studies (e.g., Friedlingstein...

The explanation relevant to data used is strange format (P5 L10-P7 L21). For example,

why is parameter's unit necessary here? most explanation of "P6 L28-P7 L9" is for the Zhao's work, not this study. After downloading the data, what did the author do as the data pre-processing? The explanation directly related to this study (P7L13-21) should be written at the start of the paragraph...etc.

P10 L19-21: The statement does not match with fig. 3. It is significant mistake.

P15L13-14: "In general, CARDAMOM was better at simulating GPP than JULES." Please present factual evidence if the statement is correct. The dataset is created with ground observations, and the empirical method is used to expand it from point to spatial data; CARDAMOM may include some significant error.

Fig. 7: As everybody knows, accuracy of the satellite observations is essentially not good at low latitude because of bad observed condition by cloud cover. The authors should represent the difference of GPP in not only low latitude but also other region. Since the evaluation data is global scale, you can do the comparison at global scale. If you keep the way to compare your results with others at just low latitude, please explain the reason.

Abstract; L6: delete "it was found that" L8: delete "fluxes" L9: delete "It was found that L9: between -> among L9-11: this sentence is not clear. L12: what is the meaning of "no impact"? Please add the quantitative interpretation.

---

## Referee Comment (RC2) · Anonymous Referee #2 · 26 Oct 2016

This paper provides a useful evaluation of JULES GPP using three different climate data sets at three resolutions. The simulated GPP is compared to three global benchmarking datasets.  The strength of the manuscript is the comparison to multiple datasets with different ways of running JULES. Sensitivity of model results to driving data (which itself is uncertain) is not well studied, so it's important to understand how robust properties like GPP are to the inputs for the model.  However, the paper is not very well organized and there are many places where I feel more detail could be given. I think it needs some substantial revisions.  I have two major comments, and several suggestions to help with the organization and readability of the manuscript.

Major Comments

1. Results:  I think the discussion of results in Section 3 needs some improvement,

with more detail on the processes behind the modelled and observed patterns in GPP. The focus of the paper is on comparing JULES to these datasets, but it would be more interesting to first explain what the datasets show. Lead each section with a brief explanation of the observed pattern in GPP, and explain differences between the datasets. Then the results of JULES can be given within the context of the observations and CARDAMOM.

- For example, in Figure 2b, JULES does very well if you are only comparing to MODIS. But it overestimates the variability of GPP during winter months compared to the other two datasets. So does this mean that JULES captures the interannual variability, or not?

- Another example: Page 11, Lines 29-33 (Discussion of figure 6): Why are the results for the extratropics the only ones discussed? I think much more could be said here – instead of just listing the differences it would be better to provide some more evaluation. For example, it was already stated that JULES overestimates GPP in the tropics, and this analysis shows that the overestimation occurs in all tropical land areas. That is a useful thing to note. On the other hand, JULES does reasonable in the extratropics – but it is consistently lower than all three datasets in Northern Asia.

2. Robustness of results: A potential strength of this manuscript is the comparison of JULES using different datasets, however I found the discussion of this topic a bit thin. Could the authors provide some more detailed discussion and context of the results? Here are some examples where further information could be provided:

- It's interesting that the results were insensitive to the spatial resolution (Page 12, Lines 9-12). This is an important conclusion of the analysis, and as the authors point out, using courser resolutions can save computational resources. But – is the result surprising given that the same soil ancillary data was used for all experiments? The JULES parameterizations are not scale-dependent (for example, this isn't the same as comparing scales in a model that uses cloud microphysical processes). I think using a

different soil ancillary data set would have a larger impact than changing the resolution.

- Also the meteorological dataset did not strongly change the results. However this is dependent on two things: 1) Maybe there were not large differences in climate between the data sets? IE: Page 15, Lines 2-6: Why are these differences in GPP occurring? Is the temperature and precipitation (or other variables) very different between the datasets in these regions? Are there other regions where the climate is very different, but the JULES simulations do not show dramatically different GPP? It would be good to provide some more information on the climates from the different driving data sets. 2) Since JULES was run with prescribed PFTs, there was no feedback between NPP and the land cover. It's possible that the GPP would be much more sensitive to the meteorology if competition between PFTs were allowed. Could the authors provide two additional experiments where the competition is allowed (e.g. one with either WFDEI product and one with the PRINCETON dataset)? Or at least provide the caveat that these results are possibly only valid when TRIFFID is not turned on. Although it's more work, I do think the additional simulations with TRIFFID would make this paper more relevant to a larger audience, as it seems most investigations using JULES have TRIFFID predicting PFTs (for example in TRENDY, the HELIX project, ISIMIP, and most CMIP5 and upcoming CMIP6 experiments).

Other Comments

1. There are several places where the text is repetitive:

- GPP is important because errors in its calculation can propagate through the model and affect biomass and other flux calculations: Page 2, Lines 27-28; Page 3, Line 5; Page 4 Lines 31-33.

- JULES is compared against FLUXNET-MTE, MODIS GPP, and CARDAMOM: Page 4, Lines 1-2; Page 5, Lines 6-7; Page 5, Line 11.

- Simulations are 2001-2010 because of availability of data: Page 4, Lines 33-34; Page

5, Lines 6-7

- The list of driving meteorological variables is given three times on pages 5-6. Even though there are differences between what is available from WATCH vs PRINCETON, this information could be given in a more concise manner.

- The FLUXNET-MTE is described as being derived from a machine learning technique/model tree ensemble twice in lines 14-20 of Page 6.

- Section 2.4

- There are more examples of this, please proofread the text and remove all repetition.

2. Page 2, Lines 6-7: It would be incorrect to say the reduced ability of land to absorb CO2 in the future has been observed. Perhaps better to say ". . . has been shown by models and inferred from observations . . ."

3. Page 3, Lines 5-9: This paragraph needs some revision. The comparison of JULES to these precise datasets is not an important part of model development in general. Would be better to say that evaluating the simulated GPP at a range of scales and its sensitivity to spatial resolution and meteorological data is essential for informing future model developments. The specific datasets can be mentioned next, ie "In this manuscript, we do this using the FLUXNET-MTE etc."

4. Page 3, Line 25: I suggest removing "In LSMs"

5. Page 5, Lines 11-12: Please specify what information is provided by the soil dataset.

6. Page 5, Lines 17-19: I don't see why the requirement for data at 6 hourly intervals or less leads to the need for a number of datasets. However, there is value in evaluating model response to a number of datasets – for example JULES is currently run with different datasets for a number of projects and MIPs, and it is not known to what extent these different datasets affect the results.

7. Page 7, Lines 10-11: What is meant by modelling "quality"?

8. All evaluation of GPP is based on area-weighted GPP, correct? I think this could be said once in Section 2.5 and then it does not need to be repeated throughout the remainder of the text.

9. I would lead the results with the evaluation of the global GPP, then examine seasonal and interannual variation (ie switch sections 3.2 and 3.1). The seasonal cycle discussion does not belong in the section on interannual variability. This section should be renamed "Seasonal and interannual variability." Each section in the results ends with a one sentence summary – consider moving this sentence to the beginning of each section instead.

10. Page 10, Lines 8-9: I would move the last sentence of this paragraph to earlier in the paragraph since it explains how the reader should interpret the CV plot.

11. Page 10, Lines 13-15: This sentence is unclear.

12. Page 10, Lines 21-23: These numbers are different from what's given in Figure 3.

13. In Section 3.3, it's a bit unusual to give total over the 10 year period, instead of annual fluxes, which is what is more usually reported in global-scale evaluations of GPP.

14. Throughout the results, it would be much easier to read through if a range of the results are given instead of listing each GPP value every time. For example, Page 11, Line 15: Replace with "JULES overestimates total annual GPP by 20-41%"

15. Page 12, Lines 22, 24: I think it would be more appropriate to refer to the "pattern" of zonal means rather than the "trend" in zonal means, as trends typically refer to change in time, rather than change in space.

16. It's difficult to distinguish between the reds and pinks in Figures 2, 3, and 5; and between the shades of blue/green in Figures 4 and 6. Could a different set of colors be used?

---

## Author Comment (AC1) · 9 Mar 2017

We thank the referee for providing a review of the manuscript and agree that the suggested changes and clarifications improve it. We have made the changes outlined below in the revised manuscript. Each item starts with the reviewer's comment followed by the changes to the manuscript. The text in blue is a re-written or new paragraph/sentence which was added to the manuscript. The page and line numbers of where changes have been made to the updated manuscript are included at the end of each reply.

**Comments**

- **General comments; I can't understand the novelty of this manuscript. I agree that the novelty is the performance confirmation of JULES. Please rethink why the authors would like to show others the original results via this manuscript. And, for all things, if the content is related to just JULES unique performance confirmation, it might not directly help the reader's scientific knowledge. At such times the authors need to improve the explanation by changing the standpoint. Please rewrite the manuscript to serve to help the readers in getting maximum benefit from what the authors revealed.**

  This study provides an evaluation of JULES at global and regional scales and provides details on which ecosystems/regions to focus on for future improvements of the model. Changes have been made to the manuscript as suggested and we have added extra text to the manuscript in order to improve and clarify it.

- **Please organize all the information of the model introduction. The authors wrote them in 1. Introduction section and 2.1 Model description. Naturally the 2.1 section should be included contents directly related to this study's discussion, and omit the explanation that had little to do with this study. For example, the authors wrote the interminable explanation for the GPP calculation method, but the reader can understand several author statements in discussion section without such knowledge; there is no explanation about spatial resolution as model structure . . . etc.**

  The model description section has been re-written in order to include only the contents directly related to this study and to explain the effect of the meteorological data on photosynthesis and thus GPP (Pages 3–4). The following text was added:

  JULES is driven by the downward shortwave and longwave radiation fluxes, rainfall and snowfall rates, surface air temperature, wind speed, surface pressure and specific humidity. The downward shortwave and longwave radiation fluxes play an important role in the surface energy balance, where the downwelling radiation fluxes must equal the outgoing fluxes of sensible heat, latent heat, ground flux, reflected shortwave radiation and upwelling thermal energy, and the calculation of photosynthesis (Best et al., 2011; Clark et al., 2011). GPP is the total C used by plants in photosynthesis at the canopy scale with potential (without water and ozone stress) leaf-level photosynthesis calculated as the smoothed minimum of three limiting rates: (1) Rubisco-limited rate (determined using surface air temperature and atmospheric $CO_2$ concentrations), (2) Light-limited rate (determined using downward radiation fluxes) and (3) Rate of transport of photosynthetic products ($C_3$ plants) and PEP-Carboxylase limitation ($C_4$ plants) (determined using surface air temperature and pressure) (Clark et al., 2011). By taking soil moisture stress into account, leaf-level photosynthesis is calculated by multiplying the potential leaf-level photosynthesis by a soil moisture factor (determined using mean soil moisture concentration in the root

zone and thus, precipitation).

In JULES, there are two options available for radiation interception and the scaling of photosynthesis from leaf-level to canopy-level: (i) big leaf approach and (ii) multi-layer approach. For all model simulations performed in this study, the multi-layer approach was used which takes into account the vertical gradient of canopy photosynthetic capacity (decreasing leaf nitrogen from top to bottom of canopy) and includes light inhibition of leaf respiration (Option 4 in Table 3 of Clark et al. (2011)). Canopy-scale fluxes are estimated to be the sum of the leaf-level fluxes in each canopy layer, scaled by leaf area. LAI is calculated for each canopy level (default number is 10), with a maximum LAI prescribed for each PFT.

- **Please more explain why the authors used different climate dataset. What of the JULES GPP estimate do the authors reveal? Why did you examine just sensitivity to each dataset? (why didn't you choose the sensitivity to each meteorological parameter?). Please add the comparison among three climate datasets into results. I can't understand the impact of climate dataset on GPP (e.g., fig. 2, 3 . . . ), because I don't know the difference of the climate dataset specific feature related to this study. Moreover, please add the explanation of the relationship between JULES and the meteorological parameters in 2.1 Model description section. It means, the reader would like to know the model structural interpretation in discussing what types of calculation approach to choose.**

Reasons for why we used different meteorological datasets to drive JULES were added to the experimental design section (Page 4, lines 22–25).

A general overview is provided of how sensitive JULES GPP is to the meteorological dataset used at global scales rather than for each meteorological variable. By analysing the models sensitivity to each meteorological dataset, different analyses of the global climate are compared and therefore a multi-factor analysis of combined changes in meteorological variables can be performed.

The sensitivity to meteorological parameters was performed in Chapter 6 of the PhD thesis of Darren Slevin(Slevin, 2016). A brief summary of this sensitivity study is provided in section 4.4 (Page 15, lines 23–32).

A simple sensitivity study of the model to changes in climate (surface (2m) air temperature, precipitation and atmospheric $CO_2$ concentrations) when simulating GPP at global and regional scales for 2000–2010 was performed in Chapter 6 of the PhD thesis of Darren Slevin(Slevin, 2016) . Only changes to one climate variable were made at a time due to complex interactions associated with multiple changes in climatic factors resulting in complex non-linear ecosystem responses which can be difficult to explain. JULES GPP was found to be sensitive to changes in all three climate variables with modelled LAI only sensitive to changes in surface air temperature (Slevin, 2016). At the regional scale, for model simulations with varying air temperature, GPP increased with increasing temperature in the extratropics, but decreased with increasing temperature in the tropics. Model simulations with varying precipitation at regional scales show the same trend as those at global scales with GPP increasing with increasing precipitation and decreasing with decreasing precipitation except for the magnitude of the effect observed.

Information on how differences in the three climate datasets (WFDEI-GPCC, WFDEI-CRU and PRINCETON) affect GPP simulations has been included in various parts of the manuscript (Page 11, lines 19–24; Page 15, lines 1–16). In the model description section

(2.1), a paragraph has been added which provides an explanation of the relationship between JULES and the meteorological parameters (Page 3, lines 25–30; Page 4, lines 1–5). This relationship between JULES and the meteorological parameters is also included in the discussion section (Page 12, line 30–Page 13, line 10; Page 15, lines 1–8; Page 15, line 33–Page 16, line 9).

- **The authors should organize first and second paragraph of "1. Introduction". The authors should integrate the two paragraphs into one. P1 L19-20: delete the sentence (Changes in atmospheric CO2 . . . ). P2 L2 and L4: "location of" → reservoir in? P2 L3: "Changes in the land surface" is not clear. P2 L7: "models and observations (Friedlingstein" → the existing studies (e.g., Friedlingstein . . .**

The first two paragraphs of the introduction have been combined into one with changes made to the text and repetitive text removed. The new paragraph follows (Page 1, line 18–Page 2, line 8).

The land surface is an important component of the climate system, provides the lower boundary for the atmosphere and exchanges energy, water and carbon (C) with the atmosphere (Pielke et al., 1998; Pitman, 2003; Seneviratne and Stöckli, 2008). It also controls the partitioning of available energy (into latent and sensible heat) and water (into evaporation and runoff) at the surface (Bonan, 2008). Changes in the land surface due to human activities, such as those from tropical deforestation, can influence climate at various time and spatial scales and since the land surface is the location of the terrestrial C cycle, it's ability to act as a C source or sink can influence atmospheric $CO_2$ concentrations (Le Quéré et al., 2009; Pan et al., 2011; Le Quéré et al., 2013; Tian et al., 2016). The reduced ability of the land surface to absorb increased anthropogenic $CO_2$ emissions in the future has been shown by models and inferred from observations (Friedlingstein et al., 2006; Canadell et al., 2007; Friedlingstein et al., 2014; Sitch et al., 2015). Friedlingstein et al. (2006) and Friedlingstein et al. (2014) have suggested that a major source of model uncertainty is the land C cycle and this can affect the ability of earth system models (ESMs; also known as coupled carbon-cycle–climate models) to reliably simulate future atmospheric $CO_2$ concentrations and climate (Dalmonech et al., 2014).

- **The explanation relevant to data used is strange format (P5 L10-P7 L21). For example, why is parameter's unit necessary here? most explanation of "P6 L28-P7 L9" is for the Zhao's work, not this study. After downloading the data, what did the author do as the data pre-processing? The explanation directly related to this study (P7 L13-21) should be written at the start of the paragraph . . . etc.**

The units of the meteorological variables used to drive JULES has been tidied up in the Data section (Section 2.3) (Page 5, lines 26–28 and lines 32–33). The information provided regarding Zhao's work has been shortened (Page 6, lines 14-26). Information on how the data was pre-processed has been included at the end of the paragraph (Page 6, lines 24-26). The paragraph regarding CARDAMOM was structured in such a way that general information on the framework was put first followed by the model output used in this study (Page 6, lines 27–34; Page 7, lines 1–4).

- **P10 L19-21: The statement does not match with fig. 3. It is significant mistake.**

The statement now reads (Page 8, lines 24–26)

This value is greater than that estimated by MODIS, FLUXNET-MTE and CARDAMOM with annual average global GPP estimated to be 112, 130 and $114 \, \mathrm{Pg \, C \, year^{-1}}$, respectively, for the same period (Figures 2a, b and d).

- **P15 L13-14: "In general, CARDAMOM was better at simulating GPP than JULES.". Please present factual evidence if the statement is correct. The dataset is created with ground observations, and the empirical method is used to expand it from point to spatial data; CARDAMOM may include some significant error.**
The statement "In general, CARDAMOM was better at simulating GPP than JULES." was used since global GPP simulated by CARDAMOM GPP (Figure 2) and the pattern of zonal means of total annual model simulated GPP (Figure 5) was between that of MODIS and FLUXNET-MTE. However, we removed this sentence from the Conclusions section (Page 17, lines 1–7). All GPP estimates have errors, but these are not always quantified and provided.

In the conclusions, the following paragraph was added which discusses the sources of error in the three benchmarking datasets (Page 17, line 30–Page 18, line 4).

The three benchmarking datasets all contain sources of error. Since observations of GPP do not exist at global scales, the MODIS and FLUXNET-MTE datasets are referred to as observation-based estimates of GPP as they are generated using observations and models. CARDAMOM may contain significant error from the assimilated data and model structure (number of pools, fire resilience of ecosystems), but so do the empirically based FLUXNET-MTE data (up-scaling of a partitioning algorithm) and MODIS GPP (a model based on PFT specific light-use efficiency). The advantage of CARDAMOM is that it is a process-based model and it ensures that the whole ecosystem functioning is coherent, while the observation-based datasets are only empirically based representations of GPP. In Figure S4 of the Supplementary Information of Bloom et al. (2016), there is a detailed study of the sensitivity of CARDAMOM to these various factors at 4 selected pixels representing temperate, boreal, wet and dry tropical ecosystems. Overall, there is not much difference in retrieved parameters because of the large error/uncertainty terms used when computing the likelihood.

- **Fig. 7: As everybody knows, accuracy of the satellite observations is essentially not good at low latitude because of bad observed condition by cloud cover. The authors should represent the difference of GPP in not only low latitude but also other region. Since the evaluation data is global scale, you can do the comparison at global scale. If you keep the way to compare your results with others at just low latitude, please explain the reason.**
The reason for only examining the difference in GPP fluxes between 15°N–30°N (Figure 5) was to find out which region contributed most to this difference and the possible reasons behind it. We suggest that this difference in GPP was due to incorrect simulation by JULES in Mexico (Figure 7). Even when JULES was driven with multiple meteorological datasets, it was unable to simulate GPP in this region (Figure 5) (Page 12, lines 14–29; Page 15, lines 6–8).

- **Abstract; L6: delete "it was found that" L8: delete "fluxes" L9: delete "It was found that L9: between → among L9-11: this sentence is not clear. L12: what is the meaning of "no impact" → Please add the quantitative interpretation.**
The words were deleted as suggested (Page 1, lines 5–7; Page 1, line 8; Page 1, lines

9–10). The sentence at lines 9-11 was re-written (Page 1, lines 9–10). A quantitative interpretation was added to Line 12 (Page 1, lines 10–12).

**Bibliography**

M. J. Best, M. Pryor, D. B. Clark, G. G. Rooney, R .L. H. Essery, C. B. Ménard, J. M. Edwards, M. A. Hendry, A. Porson, N. Gedney, L. M. Mercado, S. Sitch, E. Blyth, O. Boucher, P. M. Cox, C. S. B. Grimmond, and R. J. Harding. The Joint UK Land Environment Simulator (JULES), Model description–Part 1: Energy and water fluxes. *Geoscientific Model Development*, 4:677–699, 2011. doi: 10.5194/gmd-4-677-2011.

A. A. Bloom, J.-F. Exbrayat, I. R. van der Velde, L. Feng, and M. Williams. The decadal state of the terrestrial carbon cycle: Global retrievals of terrestrial carbon allocation, pools, and residence times. *Proceedings of the National Academy of Sciences*, 113:1285–1290, 2016. doi: 10.1073/pnas.1515160113.

G. B. Bonan. Forests and Climate Change: Forcings, Feedbacks, and the Climate Benefits of Forests. *Science*, 320:1444–1449, 2008. doi: 10.1126/science.1155121.

J. G. Canadell, C. Le Quéré, M. R. Raupach, C. B. Field, E. T. Buitenhuis, P. Ciais, T. J. Conway, N. P. Gillett, R. A. Houghton, and G. Marland. Contributions to accelerating atmospheric $CO_2$ growth from economic activity, carbon intensity, and efficiency of natural sinks. *Proceedings of the National Academy of Sciences of the United States of America*, 104: 18866–18870, 2007. doi: 10.1073/pnas.0702737104.

D. B. Clark, L. M. Mercado, S. Sitch, C. D. Jones, N. Gedney, M. J. Best, M. Pryor, G. G. Rooney, R. L. H. Essery, E. Blyth, O. Boucher, R. J. Harding, C. Huntingford, and P. M. Cox. The Joint UK Land Environment Simulator (JULES), model description–Part 2: Carbon fluxes and vegetation dynamics. *Geoscientific Model Development*, 4:701–722, 2011. doi: 10.5194/gmd-4-701-2011.

D. Dalmonech, S. Zaehle, G. J. Schürmann, V. Brovkin, C. Reick, and R. Schnur. Separation of the Effects of Land and Climate Model Errors on Simulated Contemporary Land Carbon Cycle Trends in the MPI Earth System Model version 1. *Journal of Climate*, 28:272–291, 2014. doi: 10.1175/JCLI-D-13-00593.1.

P. Friedlingstein, P. Cox, R. Betts, L. Bopp, W. Von Bloh, V. Brovkin, P. Cadule, S. Doney, M. Eby, I. Fung, et al. Climate-Carbon Cycle Feedback Analysis: Results from the C4MIP Model Intercomparison. *Journal of Climate*, 19:3337–3353, 2006. doi: 10.1175/JCLI3800.1.

P. Friedlingstein, M. Meinshausen, V. K. Arora, C. D. Jones, A. Anav, S. K. Liddicoat, and R. Knutti. Uncertainties in CMIP5 climate projections due to carbon cycle feedbacks. *Journal of Climate*, 27:511–526, 2014. doi: 10.1175/JCLI-D-12-00579.1.

C. Le Quéré, M. R. Raupach, J. G. Canadell, and G. Marland. Trends in the sources and sinks of carbon dioxide. *Nature Geoscience*, 2:831–836, 2009. doi: 10.1038/ngeo689.

C. Le Quéré, R. J. Andres, T. Boden, T. Conway, R. A. Houghton, J. I. House, G. Marland, G. P. Peters, G. R. van der Werf, A. Ahlström, R. M. Andrew, L. Bopp, J. G. Canadell, P. Ciais, S. C. Doney, C. Enright, P. Friedlingstein, C. Huntingford, A. K. Jain, C. Jourdain, E. Kato, R. F. Keeling, K. Klein Goldewijk, S. Levis, P. Levy, M. Lomas, B. Poulter, M. R. Raupach, J. Schwinger, S. Sitch, B. D. Stocker, N. Viovy, S. Zaehle, and N. Zeng. The global carbon budget 1959–2011. *Earth System Science Data*, 5:165–185, 2013. doi: 10.5194/essd-5-165-2013.

Y. Pan, R. A. Birdsey, J. Fang, R. Houghton, P. E. Kauppi, W. A. Kurz, O. L. Phillips, A. Shvidenko, S. L. Lewis, J. G. Canadell, P. Ciais, R. B. Jackson, S. W. Pacala, A. D. McGuire, S. Piao, A. Rautiainen, S. Sitch, and D. Hayes. A Large and Persistent Carbon Sink in the World's Forests. *Science*, 333:988–993, 2011. doi: 10.1126/science.1201609.

R. A. Pielke, R. Avissar, M. Raupach, A. J. Dolman, X. Zeng, and A. S. Denning. Interactions between the atmosphere and terrestrial ecosystems: influence on weather and climate. *Global Change Biology*, 4:461–475, 1998. doi: 10.1046/j.1365-2486.1998.t01-1-00176.x.

A. J. Pitman. The evolution of, and revolution in, land surface schemes designed for climate models. *International Journal of Climatology*, 23:479–510, 2003. doi: 10.1002/joc.893.

S. I. Seneviratne and R. Stöckli. *Climate Variability and Extremes during the Past 100 Years*, chapter The Role of Land-Atmosphere Interactions for Climate Variability in Europe, pages 179–193. Springer, 2008.

S. Sitch, P. Friedlingstein, N. Gruber, S. D. Jones, G. Murray-Tortarolo, A. Ahlström, S. C. Doney, H. Graven, C. Heinze, C. Huntingford, S. Levis, P. E. Levy, M. Lomas, B. Poulter, N. Viovy, S. Zaehle, N. Zeng, A. Arneth, G. Bonan, L. Bopp, J. G. Canadell, F. Chevallier, P. Ciais, R. Ellis, M. Gloor, P. Peylin, S. L. Piao, C. Le Quéré, B. Smith, Z. Zhu, and R. Myneni. Recent trends and drivers of regional sources and sinks of carbon dioxide. *Biogeosciences*, 12:653–679, 2015. doi: 10.5194/bg-12-653-2015.

D. Slevin. *Investigating sources of uncertainty associated with the JULES land surface model*. PhD thesis, School of GeoSciences, University of Edinburgh, 2016. URL `http://hdl.handle.net/1842/18757`.

H. Tian, C. Lu, P. Ciais, A. M. Michalak, J. G. Canadell, E. Saikawa, D. N. Huntzinger, K. R. Gurney, S. Sitch, B. Zhang, J. Yang, P. Bousquet, L. Bruhwiler, G. Chen, E. Dlugokencky, P. Friedlingstein, J. Melillo, S. Pan, B. Poulter, R. Prinn, M. Saunois, C. R. Schwalm, and S. C. Wofsy. The terrestrial biosphere as a net source of greenhouse gases to the atmosphere. *Nature*, 531:225–228, 2016. doi: 10.1038/nature16946.

---

## Author Comment (AC2) · 9 Mar 2017

We thank the referee for providing a review of the manuscript and agree that the suggested changes and clarifications improve it. We have made the changes outlined below in the revised manuscript. Each item starts with the reviewer's comment followed by the changes to the manuscript. The text in blue is a re-written or new paragraph/sentence which has been added to the manuscript. The page and line numbers of where changes have been made to the updated manuscript are included at the end of each reply.

**Major Comments**

1. **Results: I think the discussion of results in Section 3 needs some improvement, with more detail on the processes behind the modelled and observed patterns in GPP. The focus of the paper is on comparing JULES to these datasets, but it would be more interesting to first explain what the datasets show. Lead each section with a brief explanation of the observed pattern in GPP, and explain differences between the datasets. Then the results of JULES can be given within the context of the observations and CARDAMOM.**

   As suggested, in the discussion section, We have started each subsection with a brief explanation of the pattern in observed and CARDAMOM GPP, followed by differences between the datasets. Finally, JULES GPP is given within the context of the observations and CARDAMOM. This has been done for subsections 4.1 and 4.2 (Pages 11–14). This has not been done for subsections 4.3 and 4.4 since the performance of JULES is being evaluated against itself. An extra paragraph was added to subsection 4.3 regarding the effect of spatial resolution on GPP simulations (Page 14, lines 22–26).

   Using a different soil ancillary dataset or land cover map (which specifies the PFT fractions) may have a larger impact than changing the spatial resolution. The regridding method used in this study was the conservative method, which preserves the same information when interpolating from $0.5° \times 0.5°$ to $1° \times 1°$ and $2° \times 2°$ spatial resolutions, and results in only small differences in global GPP between the model simulations with varying spatial resolution. These small differences are due to differences in the PFT fractions of the land cover map after regridding.

2. **For example, in Figure 2b, JULES does very well if you are only comparing to MODIS. But it overestimates the variability of GPP during winter months compared to the other two datasets. So does this mean that JULES captures the interannual variability, or not?**

   In Figure 3b, JULES does very well if it is only compared to MODIS and overestimates the variability of GPP during winter months compared to the other two datasets (FLUXNET-MTE and CARDAMOM). We would say that JULES captures interannual variability since the coefficient of variation (CV) expressed as percentages of the mean monthly GPP for JULES lies between the CV values for the three observation-based estimates (Page 12, lines 3–5).

3. **Another example: Page 11, Lines 29–33 (Discussion of figure 6): Why are the results for the extratropics the only ones discussed? I think much more could be said here - instead of just listing the differences it would be better to provide some more evaluation. For example, it was already stated that JULES overestimates GPP in the tropics, and this analysis shows that the overestimation occurs in all tropical land areas. That is a useful thing to note.**

**On the other hand, JULES does reasonable in the extratropics - but it is consistently lower than all three datasets in Northern Asia.**

The results for the tropics has been added including some suggestions for improving simulated GPP. The following paragraphs were added to sections 3.3 and 4.2, respectively (Page 10, lines 15–21; Page 13, lines 17–19).

JULES overestimates GPP in all three tropical land areas compared to MODIS, FLUXNET-MTE and CARDAMOM (Figures 6c, e and f). Differences between JULES, MODIS, FLUXNET-MTE and CARDAMOM GPP with average annual GPP range from 7.4–12.1 $\mathrm{Pg\,C\,year^{-1}}$, 7.7–13 $\mathrm{Pg\,C\,year^{-1}}$ and 1–1.3 $\mathrm{Pg\,C\,year^{-1}}$ for forests, grasslands and shrubs, respectively, in South and South-East Asia, 9.5–13.7 $\mathrm{Pg\,C\,year^{-1}}$, 8.4–12.3 $\mathrm{Pg\,C\,year^{-1}}$ and 1.7–2.1 $\mathrm{Pg\,C\,year^{-1}}$ for forests, grasslands and shrubs, respectively, in Africa and 18-23.2 $\mathrm{Pg\,C\,year^{-1}}$, 9–12.9 $\mathrm{Pg\,C\,year^{-1}}$ and 1.4–1.8 $\mathrm{Pg\,C\,year^{-1}}$ for forests, grasslands and shrubs, respectively, in Central and South America (Figures 6c, e and f, respectively).

JULES simulated average annual GPP to be 61, 54 and 7 $\mathrm{Pg\,C\,year^{-1}}$ for forests, grasslands and shrubs, respectively. JULES (JULES-WFDEI-GPCC) simulates higher GPP than MODIS, FLUXNET-MTE and CARDAMOM at global scales and this was found to be due to higher GPP simulated by JULES for forests and grasslands in the tropics (Figure 4b).

Yes we found that JULES performs reasonably well in the extratropics (Europe, Northern Asia, North America and Greenland and the Extratropical Southern Hemisphere), with the exception of Northern Asia and North America and Greenland, where the model is either equal to or lower than all three datasets. This may be due to the inability of this version of JULES to accurately simulate GPP in boreal regions where permafrost exists. It may also due to a different land cover map being used by JULES, MODIS and FLUXNET-MTE. The following paragraph were added to section 4.2 (Page 14, lines 4–8).

In the four extratropical regions (Europe, Northern Asia, Extratropical Southern Hemisphere and North America and Greenland), JULES simulated similar GPP to MODIS, FLUXNET-MTE and CARDAMOM for the three biomes in Europe and the Extratropical Southern Hemisphere (Figures 6a and d), with the exception of Northern Asia and North America and Greenland, where the model is either equal to or lower than all three datasets (Figures 6b and g). This is due to the inability of this version of JULES to accurately simulate GPP in boreal regions where permafrost exists.

4. **Robustness of results: A potential strength of this manuscript is the comparison of JULES using different datasets, however I found the discussion of this topic a bit thin. Could the authors provide some more detailed discussion and context of the results? Here are some examples where further information could be provided: It's interesting that the results were insensitive to the spatial resolution (Page 12, Lines 9-12). This is an important conclusion of the analysis, and as the authors point out, using courser resolutions can save computational resources. But is the result surprising given that the same soil ancillary data was used for all experiments? The JULES parameterizations are not scale-dependent (for example, this isn't the same as comparing scales in a model that uses cloud microphysical processes). I think using a different soil ancillary data set would have a larger impact than changing the resolution.**

Yes, we found it interesting that the results were insensitive to spatial resolution. This is a useful since lower resolution global simulations can be performed to save computational resources. Using a different soil ancillary dataset or land cover map would have a larger

impact than changing the resolution. A paragraph discussing these points has been added to section 4.3 (Page 14, line 22–26).

*Using a different soil ancillary dataset or land cover map (which specifies the PFT fractions) may have a larger impact than changing the spatial resolution. The regridding method used in this study was the conservative method, which preserves the same information when interpolating from 0.5° × 0.5° to 1° × 1° and 2° × 2° spatial resolutions, and results in only small differences in global GPP between the model simulations with varying spatial resolution. These small differences are due to differences in the PFT fractions of the land cover map after regridding.*

5. **Also the meteorological dataset did not strongly change the results. However this is dependent on two things: 1) Maybe there were not large differences in climate between the data sets? IE: Page 15, Lines 2-6: Why are these differences in GPP occurring? Is the temperature and precipitation (or other variables) very different between the datasets in these regions? Are there other regions where the climate is very different, but the JULES simulations do not show dramatically different GPP? It would be good to provide some more information on the climates from the different driving data sets. 2) Since JULES was run with prescribed PFTs, there was no feedback between NPP and the land cover. It's possible that the GPP would be much more sensitive to the meteorology if competition between PFTs were allowed. Could the authors provide two additional experiments where the competition is allowed (e.g. one with either WFDEI product and one with the PRINCETON dataset)? Or at least provide the caveat that these results are possibly only valid when TRIFFID is not turned on. Although it's more work, I do think the additional simulations with TRIFFID would make this paper more relevant to a larger audience, as it seems most investigations using JULES have TRIFFID predicting PFTs (for example in TRENDY, the HELIX project, ISIMIP, and most CMIP5 and upcoming CMIP6 experiments).**

1) When JULES was driven with different meteorological datasets, differences in simulated GPP occurred mostly in the tropics (between 5°N-5°S) with JULES driven with WFDEI-GPCC-1degree simulating higher GPP than JULES driven with PRINCETON and slightly higher GPP in the extratropics was simulated by JULES was driven with PRINCETON (Figure 5). There are differences in climate between the two datasets. Positive biases in the downward longwave radiation fluxes and surface air temperatures in the meteorological datasets are the reason for these differences (Figures G.5 and G.6 in Slevin (2016)). In general, precipitation in the WFDEI-GPCC dataset is higher than that of PRINCETON (Figures G.6b and d in Slevin (2016)) with surface air temperatures higher in PRINCETON (Figures G.6a and c in Slevin (2016)). However, since JULES is more sensitive to downward longwave radiation and surface air temperature than precipitation when simulating GPP, the main reason for differences in simulated GPP when JULES was driven with two different meteorological datasets is due to differences in downward longwave radiation fluxes and surface air temperatures. There are differences in northern Eurasia (above 60°N) in the meteorological datasets with slightly higher radiation fluxes (downward shortwave and longwave) and surface air temperatures in the PRINCETON dataset with little difference between the JULES simulations driven with WFDEI-GPCC and PRINCETON in this region (Figure 5). Information on differences in the meteorological dataset (WFDEI-GPCC and PRINCETON) led to differences in simulated GPP has been added to section 4.4 on Page 15, lines 1–16.

The higher simulated GPP in the tropics when JULES was driven with WFDEI-GPCC is due to positive biases in downward longwave radiation fluxes in WFDEI-GPCC in the Amazonian, African and South-East Asian tropics (Figures G.5b and d in Slevin (2016)) and the higher GPP simulated by JULES (driven with PRINCETON) in the extratropics are a result of positive biases in downward longwave radiation in the PRINCETON dataset in North America and Northern Asia (Figure G.5b in Slevin (2016)) and positive biases in surface air temperature in the PRINCETON dataset in the Northern Hemisphere (Figures G.6a and c in Slevin (2016)). As with the JULES-WFDEI-GPCC simulations, there are also differences in GPP between the PRINCETON driven JULES simulation and the observation-based and CARDAMOM estimates at latitudes 15°N-30°N (Figure 5). There was no improvement in simulated GPP when a different meteorological dataset was used.

In general, precipitation in the WFDEI-GPCC dataset is higher than that of PRINCETON (Figures G.6b and d in Slevin (2016)) with surface air temperatures higher in PRINCETON (Figures G.6a and c in Slevin (2016)). However, since JULES is more sensitive to downward longwave radiation and surface air temperature than precipitation when simulating GPP(Alton et al., 2007), the main reason for differences in simulated GPP when JULES was driven with two different meteorological datasets is due to differences in downward longwave radiation fluxes and surface air temperatures. There are differences in northern Eurasia (above 60°N) in the meteorological datasets with slightly higher radiation fluxes (downward shortwave and longwave) and surface air temperatures in the PRINCETON dataset with little difference between the JULES simulations driven with WFDEI-GPCC and PRINCETON in this region (Figure 5).

and on Page 15, line 33–Page 16, line 9.

When JULES was driven with the PRINCETON dataset, it was found that simulated photosynthesis was mostly Rubisco-limited (Figure 5.25 in Slevin (2016)). A similar trend was found when JULES was driven with the WFDEI-GPCC dataset (Figure 5.6 in Slevin (2016)). Similar trends in transport limitation were found with the JULES-PRINCETON model simulation, though the number of model gridboxes in which transport limitation dominated was less than that for the JULES-WFDEI-GPCC-1degree model simulation (Figures 5.25 and 5.28 in Slevin (2016)). When comparing the model gridbox fractions for the JULES-WFDEI-GPCC-1degree and JULES-PRINCETON model simulations, it was found that when JULES was driven with the PRINCETON dataset, simulated photosynthesis was more Rubisco-limited than when the model was driven with WFDEI-GPCC (Figure 5.26 in Slevin (2016)). Light-limitation was more important in simulating photosynthesis when JULES was driven with WFDEI-GPCC than PRINCETON (Figure 5.27 in Slevin (2016)). The percentage of model gridboxes which are transport-limited show a pronounced geographical variation with the WFDEI-GPCC driven simulation being more transport-limited in the Southern Hemisphere and the PRINCETON driven simulation being more transport-limited in the Northern Hemisphere (Figure 5.28 in Slevin (2016)).

2) Yes, since JULES was run with prescribed PFTs, there was no feedback between NPP and the land cover and there is a possibility that GPP could be more sensitive to the meteorology if competition between PFTs were allowed. These additional simulations with TRIFFID would make this paper more relevant to a larger audience. Therefore, two more model simulations were carried out where vegetation competition (and TRIFFID) were switched on. This was done with the WFDEI-GPCC and PRINCETON datasets (both at $1° \times 1°$ spatial resolution). A new figure was added showing the results from these extra model simulations (Page 32, Figure 8). A paragraph describing the results

from these simulations was added to section 4.4 (Page 16, lines 10-24)

In this study, the model simulations were performed with prescribed PFTs (i.e. no vegetation competition). If competition between PFTs was allowed (i.e. vegetation competition), the annual average global GPP would be higher by 15 % and 17 %, for the WFDEI-GPCC and PRINCETON driven simulations, respectively (Figures 8b and e). In general, with vegetation competition switched on, higher GPP was simulated by JULES when driven with both datasets (Figures 8c and f). Higher GPP occurred mostly in Europe, southeastern US, and in the tropical regions of Central and South America, Africa and South and South-East Asia (Figures 8c and f). This increased GPP in tropical regions is due to the tree-shrub-grass dominance heirachy in TRIFFID with dominant types (trees) limiting the expansion of subdominant types (shrubs and grasses). In savanna regions, such as the Sudanian Savanna, which stretches from the Atlantic Ocean in the west to the Ethiopian Highlands in the east of Africa, and northern Australia, there is higher GPP with prescribed PFTs (Figures 8c and f). These are also fire-prone regions. The version of JULES used in this study has no fire module and TRIFFID may overestimate woody cover and therefore GPP.

In terms of global GPP, the WFDEI-GPCC and PRINCETON driven simulations produce similar increases (Figures 8b and e). However, the spatial pattern is slightly different with higher GPP simulated in the Amazon region when JULES was driven with the WFDEI-GPCC dataset and higher GPP in southern Brazil and Argentina and Southeast Asia when JULES was driven with the PRINCETON dataset (Figures 8c and f). The spatial pattern of simulated GPP is more sensitive to the meteorological data than the annual average global GPP if competition between PFTs is allowed. This may be due to compensating differences in the sensitivity of the model to the two meteorological datasets.

and to the conclusions (Page 17, lines 26–29).

The model simulations in this study were largely performed with prescribed PFTs (i.e. no competition between PFTs was allowed). With competition between PFTs, the annual average global GPP was higher by 15 % and 17 %, for the WFDEI-GPCC and PRINCETON driven simulations, respectively, with the spatial pattern of simulated GPP more sensitive to the meteorological data used.

**Other Comments**

1. **There are several places where the text is repetitive:**
   The text has been updated to avoid repetition (see below).

   - **GPP is important because errors in its calculation can propagate through the model and affect biomass and other flux calculations: Page 2, Lines 27–28; Page 3, Line 5; Page 4 Lines 31–33.**
     Done (Page 2, lines 19–24).

   - **JULES is compared against FLUXNET-MTE, MODIS GPP, and CARDAMOM: Page 5, Lines 1–2; Page 5, Lines 6–7; Page 5, Line 11.**
     Done (Page 4, lines 27–28).

   - **Simulations are 2001–2010 because of availability of data: Page 4, Lines 33–34; Page 5, Lines 6–7.**
     Done (Page 4, lines 26–27).

- **The list of driving meteorological variables is given three times on pages 5–6. Even though there are differences between what is available from WATCH vs PRINCETON, this information could be given in a more concise manner.**
  Done (Page 5, lines 26–28 and 32–33). The driving meteorological variables is also listed in the model description section since it is required when explaining the connection between the meteorological variables and GPP (Page 3, lines 25–26).

- **The FLUXNET-MTE is described as being derived from a machine learning technique/ model tree ensemble twice in lines 14–20 of Page 6.**
  Done (Page 6, lines 2–6).

- **Section 2.4 - There are more examples of this, please proofread the text and remove all repetition.**
  This section was re-written in order to avoid repetition (Page 7, lines 5–29).

2. **Page 2, Lines 6–7: It would be incorrect to say the reduced ability of land to absorb CO2 in the future has been observed. Perhaps better to say "...has been shown by models and inferred from observations ..."**
  This sentence has been changed to (Page 2, lines 3–5)

  The reduced ability of the land surface to absorb increased anthropogenic $CO_2$ emissions in the future has been shown by models and inferred from observations (Friedlingstein et al., 2006; Canadell et al., 2007; Friedlingstein et al., 2014; Sitch et al., 2015).

3. **Page 3, Lines 5–9: This paragraph needs some revision. The comparison of JULES to these precise datasets is not an important part of model development in general. Would be better to say that evaluating the simulated GPP at a range of scales and its sensitivity to spatial resolution and meteorological data is essential for informing future model developments. The specific datasets can be mentioned next, ie "In this manuscript, we do this using the FLUXNET-MTE etc."**
  This paragraph has been re-written (Page 2, line 33–Page 3, line 2).

  JULES has been evaluated at various scales: point (Blyth et al., 2010, 2011; Slevin et al., 2015; Ménard et al., 2015), regional (Galbraith et al., 2010; Burke et al., 2013; Chadburn et al., 2015) and globally as part of model-intercomparison studies (Anav et al., 2015; Sitch et al., 2015). Evaluating simulated GPP at a range of scales and its sensitivity to spatial resolution and meteorological data is essential for informing future model developments. In this manuscript, we do this using two observation-based datasets (FLUXNET-MTE and MODIS) and the CARbon DAta MOdel fraMework (Bloom et al., 2016, CARDAMOM).

4. **Page 3, Line 25: I suggest removing "In LSMs"**
  Done (Page 3, lines 18–19).

5. **Page 5, Lines 11–12: Please specify what information is provided by the soil dataset.**
  The following information is provided by the soil dataset (Page 5, lines 3–6).

  The soil dataset used was the Harmonized World Soil Database version 1.2 (Nachtergaele et al., 2012, HWSD) and contains soil property data such as soil texture fractions, water storage capacity, soil depth and pH (Nachtergaele et al., 2012). In this study, the soil texture fractions (% of sand, silt and clay) were used to calculate the soil thermal and hydraulic conductivity parameters listed in Table 3 of Best et al. (2011).

6. **Page 5, Lines 17-19: I don't see why the requirement for data at 6 hourly intervals or less leads to the need for a number of datasets. However, there is value in evaluating model response to a number of datasets - for example JULES is currently run with different datasets for a number of projects and MIPs, and it is not known to what extent these different datasets affect the results.**

   Yes, I agree that evaluating JULES' response to various datasets (soil, vegetation and meteorological) can help to explain its behaviour when used as part of a multi-model inter-comparison project. This sentence has been changed to (Page 5, lines 8–10)

   Two meteorological datasets were used to drive the model offline (i.e. run separately from its host Earth System Model) at global scales; WFDEI (Weedon et al., 2014) and PRINCE-TON (Sheffield et al., 2006).

7. **Page 7, Lines 10-11: What is meant by modelling "quality"?**

   The sentence is missing the word "improve". It now reads (Page 6, lines 27–29)

   The CARbon DAta MOdel fraMework (CARDAMOM) is a model-data fusion approach which consists of merging observational data with models in order to improve model quality and characterise its uncertainty.

8. **All evaluation of GPP is based on area-weighted GPP, correct? I think this could be said once in Section 2.5 and then it does not need to be repeated throughout the remainder of the text.**

   Yes, all evaluation of GPP is based on area-weighted GPP. Since it is mentioned in Section 2.5, it has been removed from the remainder of the text.

9. **I would lead the results with the evaluation of the global GPP, then examine seasonal and interannual variation (ie switch sections 3.2 and 3.1). The seasonal cycle discussion does not belong in the section on interannual variability. This section should be renamed "Seasonal and interannual variability." Each section in the results ends with a one sentence summary - consider moving this sentence to the beginning of each section instead.**

   Sections 3.2 and 3.1 were switched (Pages 8–9). This also required that the abstract (Page 1, lines 5–8), the study questions (Page 3, lines 6–10), the list of experiments (Section 2.4; Page 7) and parts of Section 4.1 (Discussion; Page 12–14) and the Conclusions (Section 5; Page 16–18) be slightly re-written. Section 3.2 has been renamed to "Seasonal and interannual variability of GPP." (Page 9). The one sentence summary at the end of each section in the results section has been moved to the beginning.

10. **Page 10, Lines 8-9: I would move the last sentence of this paragraph to earlier in the paragraph since it explains how the reader should interpret the CV plot.**

    The last sentence of this paragraph has been moved to earlier in the paragraph as suggested (Page 9, lines 13–15).

11. **Page 10, Lines 13-15: This sentence is unclear.**

    This sentence has been rewritten (Page 9, lines 22–23).

    The model is able to capture simulated monthly anomalies from 2001 to 2010 with the exception of those in 2002 (Figure 3c).

12. **Page 10, Lines 21-23: These numbers are different from what's given in Figure 3.**
The numbers have been changed to reflect those given in Figure 2 (Page 8, lines 24–27).

This value is greater than that estimated by MODIS, FLUXNET-MTE and CARDAMOM with annual average global GPP estimated to be 112, 130 and $114 \, \text{Pg} \, \text{C} \, \text{year}^{-1}$, respectively, for the same period (Figures 2a, b and d). The higher global GPP simulated by the JULES-WFDEI-GPCC driven simulations is greater than the MODIS, FLUXNET-MTE and CARDAMOM estimates by $25\,\%$, $8\,\%$ and $23\,\%$ on average, respectively.

13. **In Section 3.3, it's a bit unusual to give total over the 10 year period, instead of annual fluxes, which is what is more usually reported in global-scale evaluations of GPP.**
Annual fluxes have been provided for GPP in Section 3.3 (Pages 9–10).

14. **Throughout the results, it would be much easier to read through if a range of the results are given instead of listing each GPP value every time. For example, Page 11, Line 15: Replace with "JULES overestimates total annual GPP by 20-41%"**
The results section has now been changed so that a range of results are given instead of listing each GPP value every time (Pages 8–11).

15. **Page 12, Lines 22, 24: I think it would be more appropriate to refer to the "pattern" of zonal means rather than the "trend" in zonal means, as trends typically refer to change in time, rather than change in space.**
Yes, you are correct. This has been changed (Page 11, lines 15–18).

16. **It's difficult to distinguish between the reds and pinks in Figures 2, 3, and 5; and between the shades of blue/green in Figures 4 and 6. Could a different set of colors be used?**
A different set of colors has been used to distinguish between the various model simulations in order to make it easier for the reader (Pages 26–30).

**Bibliography**

P. Alton, L. Mercado, and P. North. A sensitivity analysis of the land-surface scheme JULES conducted for three forest biomes: Biophysical parameters, model processes, and meteorological driving data. *Global Biogeochemical Cycles*, 20:GB1008, 2007. doi: 10.1029/2005GB002653.

A. Anav, P. Friedlingstein, C. Beer, P. Ciais, A. Harper, C. Jones, G. Murray-Tortarolo, D. Papale, N. C. Parazoo, P. Peylin, S. Piao, S. Sitch, N. Viovy, A. Wiltshire, and M. Zhao. Spatio-temporal patterns of terrestrial gross primary production: A review. *Reviews of Geophysics*, 2015. doi: 10.1002/2015RG000483.

M. J. Best, M. Pryor, D. B. Clark, G. G. Rooney, R .L. H. Essery, C. B. Ménard, J. M. Edwards, M. A. Hendry, A. Porson, N. Gedney, L. M. Mercado, S. Sitch, E. Blyth, O. Boucher, P. M. Cox, C. S. B. Grimmond, and R. J. Harding. The Joint UK Land Environment Simulator (JULES), Model description–Part 1: Energy and water fluxes. *Geoscientific Model Development*, 4:677–699, 2011. doi: 10.5194/gmd-4-677-2011.

A. A. Bloom, J.-F. Exbrayat, I. R. van der Velde, L. Feng, and M. Williams. The decadal state of the terrestrial carbon cycle: Global retrievals of terrestrial carbon allocation, pools, and residence times. *Proceedings of the National Academy of Sciences*, 113:1285–1290, 2016. doi: 10.1073/pnas.1515160113.

E. Blyth, J. Gash, A. Lloyd, M. Pryor, G. P. Weedon, and J. Shuttleworth. Evaluating the JULES Land Surface Model Energy Fluxes Using FLUXNET Data. *Journal of Hydrometeorology*, 11:509–519, 2010. doi: 10.1175/2009JHM1183.1.

E. Blyth, D. B. Clark, R. Ellis, C. Huntingford, S. Los, M. Pryor, M. Best, and S. Sitch. A comprehensive set of benchmark tests for a land surface model of simultaneous fluxes of water and carbon at both the global and seasonal scale. *Geoscientific Model Development*, 4:255–269, 2011. doi: 10.5194/gmd-4-255-2011.

E. J. Burke, R. Dankers, C. D. Jones, and A. J. Wiltshire. A retrospective analysis of pan Arctic permafrost using the JULES land surface model. *Climate Dynamics*, 41:1025–1038, 2013. doi: 10.1007/s00382-012-1648-x.

J. G. Canadell, C. Le Quéré, M. R. Raupach, C. B. Field, E. T. Buitenhuis, P. Ciais, T. J. Conway, N. P. Gillett, R. A. Houghton, and G. Marland. Contributions to accelerating atmospheric $CO_2$ growth from economic activity, carbon intensity, and efficiency of natural sinks. *Proceedings of the National Academy of Sciences of the United States of America*, 104:18866–18870, 2007. doi: 10.1073/pnas.0702737104.

S. Chadburn, E. Burke, R. Essery, J. Boike, M. Langer, M. Heikenfeld, P. Cox, and P. Friedlingstein. An improved representation of physical permafrost dynamics in the JULES land-surface model. *Geoscientific Model Development*, 8:1493–1508, 2015. doi: 10.5194/gmd-8-1493-2015.

P. Friedlingstein, P. Cox, R. Betts, L. Bopp, W. Von Bloh, V. Brovkin, P. Cadule, S. Doney, M. Eby, I. Fung, et al. Climate-Carbon Cycle Feedback Analysis: Results from the C4MIP Model Intercomparison. *Journal of Climate*, 19:3337–3353, 2006. doi: 10.1175/JCLI3800.1.

P. Friedlingstein, M. Meinshausen, V. K. Arora, C. D. Jones, A. Anav, S. K. Liddicoat, and R. Knutti. Uncertainties in CMIP5 climate projections due to carbon cycle feedbacks. *Journal of Climate*, 27:511–526, 2014. doi: 10.1175/JCLI-D-12-00579.1.

D. Galbraith, P. E. Levy, S. Sitch, C. Huntingford, P. Cox, M. Williams, and P. Meir. Multiple mechanisms of Amazonian forest biomass losses in three dynamic global vegetation models under climate change. *New Phytologist*, 187:647–665, 2010. doi: 10.1111/j.1469-8137.2010.03350.x.

C. B. Ménard, J. Ikonen, K. Rautiainen, M. Aurela, A. N. Arslan, and J. Pulliainen. Effects of Meteorological and Ancillary Data, Temporal Averaging, and Evaluation Methods on Model Performance and Uncertainty in a Land Surface Model. *Journal of Hydrometeorology*, 16: 2559–2576, 2015. doi: 10.1175/JHM-D-15-0013.1.

F. Nachtergaele, H. van Velthuizen, L. Verelst, D. Wiberg, N. Batjes, K. Dijkshoorn, V. van Engelen, G. Fischer, A. Jones, L. Montanarella, M. Petri, S. Prieler, E. Teixeira, and X. Shi. Harmonized World Soil Database v1.2. Technical report, International Institute for Applied Systems Analysis (IIASA), Food and Agriculture Organization of the United Nations (FAO), February 2012.

J. Sheffield, G. Goteti, and E. F. Wood. Development of a 50-year high-resolution global dataset of meteorological forcings for land surface modeling. *Journal of Climate*, 19:3088–3111, 2006. doi: 10.1175/JCLI3790.1.

S. Sitch, P. Friedlingstein, N. Gruber, S. D. Jones, G. Murray-Tortarolo, A. Ahlström, S. C. Doney, H. Graven, C. Heinze, C. Huntingford, S. Levis, P. E. Levy, M. Lomas, B. Poulter, N. Viovy, S. Zaehle, N. Zeng, A. Arneth, G. Bonan, L. Bopp, J. G. Canadell, F. Chevallier, P. Ciais, R. Ellis, M. Gloor, P. Peylin, S. L. Piao, C. Le Quéré, B. Smith, Z. Zhu, and R. Myneni. Recent trends and drivers of regional sources and sinks of carbon dioxide. *Biogeosciences*, 12:653–679, 2015. doi: 10.5194/bg-12-653-2015.

D. Slevin. *Investigating sources of uncertainty associated with the JULES land surface model.* PhD thesis, School of GeoSciences, University of Edinburgh, 2016. URL `http://hdl.handle.net/1842/18757`.

D. Slevin, S. F. B. Tett, and M. Williams. Multi-site evaluation of the JULES land surface model using global and local data. *Geoscientific Model Development*, 8:295–316, 2015. doi: 10.5194/gmd-8-295-2015.

G. P. Weedon, G. Balsamo, N. Bellouin, S. Gomes, M. J. Best, and P. Viterbo. The WFDEI meteorological forcing data set: WATCH Forcing Data methodology applied to ERA-Interim reanalysis data. *Water Resources Research*, 50:7505–7514, 2014. doi: 10.1002/2014WR015638.

---

## Author Comment (AC3) · 9 Mar 2017

[revised manuscript text omitted]

shrub, which would improve GPP simulations in the tropics (Harper et al., 2016). Improved simulation of LAI in tropical regions would also aid in reducing differences between model simulated and observation-based estimates of GPP in these regions.

In the four extratropical regions (Europe, Northern Asia, Extratropical Southern Hemisphere and North America and Green-
5  land), JULES simulated similar GPP to MODIS, FLUXNET-MTE and CARDAMOM for the three biomes in Europe and the Extratropical Southern Hemisphere (Figures 6a and d), with the exception of Northern Asia and North America and Greenland, where the model is either equal to or lower than all three datasets (Figures 6b and g). This is probably due to the inability of this version of JULES to accurately simulate GPP in boreal regions where permafrost exists.

**4.3   *How sensitive are fluxes of GPP to the spatial resolution of the model?**

10  JULES was insensitive to spatial resolution with average annual global GPP being $140\,\mathrm{Pg\,C\,year}^{-1}$, $141\,\mathrm{Pg\,C\,year}^{-1}$ and $142\,\mathrm{Pg\,C\,year}^{-1}$ at $0.5° \times 0.5°$, $1° \times 1°$ and $2° \times 2°$ spatial resolutions, respectively. This pattern was also observed in the zonal mean of total annual GPP (Figure 5). The insensitivity of the model to spatial resolution at the global scale was also observed at the regional scale when comparing simulated GPP fluxes for forests, grasslands and shrubs in the tropics and extratropics (Figure 6).

15  Little research has been performed on the effects of spatial resolution on JULES simulations (as well as other LSMs). Studies using atmospheric chemistry models have shown that the spatial resolution of the input meteorological data can affect model output (Ito et al., 2009; Pugh et al., 2013; Schaap et al., 2015). The results found here agree with those from Compton and Best (2011). Compton and Best (2011) showed that JULES was insensitive to spatial resolution when the WFD dataset was regridded from half-degree to 1-degree and 2-degree when simulating the terrestrial hydrological cycle. It was found that
20  spatial resolution had little or no effect on simulations of global mean total evaporation and total runoff. However, the study showed that JULES was sensitive to temporal resolution when simulating the same hydrological components.

Using a different soil ancillary dataset or land cover map (which specifies the PFT fractions) may have a larger impact than changing the spatial resolution. The regridding method used in this study was the conservative method, which preserves the same information when interpolating from $0.5° \times 0.5°$ to $1° \times 1°$ and $2° \times 2°$ spatial resolutions, and results in only small
25  differences in global GPP between the model simulations with varying spatial resolution. These small differences are due to differences in the PFT fractions of the land cover map after regridding.

**4.4   *Is the meteorological dataset used to drive the model important at the global scale?**

When JULES was driven with the PRINCETON dataset at $1° \times 1°$ spatial resolution (Table 2, the annual average global GPP was slightly higher by $3\,\mathrm{Pg\,C\,year}^{-1}$ than that simulated by JULES when driven with the WFDEI-GPCC dataset at
30  the same resolution. In general, differences in GPP fluxes for model simulations driven using WFDEI-GPCC and PRINCE-TON are mainly in the deep tropics (at 5°N–5°S) with JULES-WFDEI-GPCC-1degree simulating higher GPP than JULES-PRINCETON and in the extratropics at 30°N–60°N, JULES-PRINCETON simulates slightly higher GPP (Figures 5 and 6).

The higher simulated GPP in the tropics when JULES was driven with WFDEI-GPCC is due to positive biases in downward longwave radiation fluxes in WFDEI-GPCC in the Amazonian, African and South-East Asian tropics (Figures G.5b and d in Slevin (2016)) and the higher GPP simulated by JULES (driven with PRINCETON) in the extratropics are a result of positive biases in downward longwave radiation in the PRINCETON dataset in North America and Northern Asia (Figure G.5b in Slevin (2016)) and positive biases in surface air temperature in the PRINCETON dataset in the Northern Hemisphere (Figures G.6a and c in Slevin (2016)). As with the JULES-WFDEI-GPCC simulations, there are also differences in GPP between the PRINCETON driven JULES simulation and the observation-based and CARDAMOM estimates at latitudes 15°N-30°N (Figure 5). There was no improvement in simulated GPP when a different meteorological dataset was used.

In general, precipitation in the WFDEI-GPCC dataset is higher than that of PRINCETON (Figures G.6b and d in Slevin (2016)) with surface air temperatures higher in PRINCETON (Figures G.6a and c in Slevin (2016)). However, since JULES is more sensitive to downward longwave radiation and surface air temperature than precipitation when simulating GPP(Alton et al., 2007), the main reason for differences in simulated GPP when JULES was driven with two different meteorological datasets is due to differences in downward longwave radiation fluxes and surface air temperatures. There are differences in northern Eurasia (above 60°N) in the meteorological datasets with slightly higher radiation fluxes (downward shortwave and longwave) and surface air temperatures in the PRINCETON dataset with little difference between the JULES simulations driven with WFDEI-GPCC and PRINCETON in this region (Figure 5).

Other studies have shown that the meteorological dataset used to drive LSMs is a large source of uncertainty in global land surface modelling (Hicke, 2005; Jung et al., 2007; Poulter et al., 2011). Different methods are used to create time series of global gridded climate data in order to drive LSMs and this can introduce uncertainty that can propagate through model simulations (Zhao et al., 2006). Even at the point scale, differences in simulated GPP were observed when driving JULES with the WFDEI-GPCC and PRINCETON datasets (Slevin et al., 2015). As in this study, it also occurred in the tropics. The choice of meteorological dataset used to drive JULES has an important influence on GPP simulations.

A simple sensitivity study of the model to changes in climate (surface (2m) air temperature, precipitation and atmospheric $CO_2$ concentrations) when simulating GPP at global and regional scales for 2000–2010 was performed(Slevin, 2016). Only changes to one climate variable were made at a time due to complex interactions associated with multiple changes in climatic factors resulting in complex non-linear ecosystem responses which can be difficult to explain. JULES GPP was found to be sensitive to changes in all three climate variables with modelled LAI only sensitive to changes in surface air temperature (Slevin, 2016). At the regional scale, for model simulations with varying air temperature, GPP increased with increasing temperature in the extratropics, but decreased with increasing temperature in the tropics. Model simulations with varying precipitation at regional scales show the same trend as those at global scales with GPP increasing with increasing precipitation and decreasing with decreasing precipitation except for the magnitude of the effect observed. More detailed information on the sensitivity study is provided in Chapter 6 of the PhD thesis of Darren Slevin(Slevin, 2016).

When JULES was driven with the PRINCETON dataset, it was found that simulated photosynthesis was mostly Rubisco-limited (Figure 5.25 in Slevin (2016)). A similar trend was found when JULES was driven with the WFDEI-GPCC dataset (Figure 5.6 in Slevin (2016)). Similar trends in transport limitation were found with the JULES-PRINCETON model simulation, though the number of model gridboxes in which transport limitation dominated was less than that for the JULES-WFDEI-GPCC-1degree model simulation (Figures 5.25 and 5.28 in Slevin (2016)). When comparing the model gridbox fractions for the JULES-WFDEI-GPCC-1degree and JULES-PRINCETON model simulations, it was found that when JULES was driven with the PRINCETON dataset, simulated photosynthesis was more Rubisco-limited than when the model was driven with WFDEI-GPCC (Figure 5.26 in Slevin (2016)). Light-limitation was more important in simulating photosynthesis when JULES was driven with WFDEI-GPCC than PRINCETON (Figure 5.27 in Slevin (2016)). The percentage of model gridboxes which are transport-limited show 
[revised manuscript text omitted]

---

## Referee Report (RR1)

Second review of Global Evaluation of Gross Primary Productivity in the JULES Land Surface Model by D. Slevin et al.

Overall the authors have addressed most of the points I raised in my initial review. I had two major comments which the authors responded to in detail. I am satisfied with most responses but below are two further questions based on these responses.

1. [My original comment] (Discussion of figure 6): Why are the results for the extratropics the only ones discussed? I think much more could be said here - instead of just listing the differences it would be better to provide some more evaluation.

[Author Response] Yes we found that JULES performs reasonably well in the extratropics (Europe, Northern Asia, North America and Greenland and the Extratropical Southern Hemisphere), with the exception of Northern Asia and North America and Greenland, where the model is either equal to or lower than all three datasets. This may be due to the inability of this version of JULES to accurately simulate GPP in boreal regions where permafrost exists. It may also due to a different land cover map being used by JULES, MODIS and FLUXNET-MTE. The following paragraph were added to section 4.2 (Page 14, lines 4–8).

In the four extratropical regions (Europe, Northern Asia, Extratropical Southern Hemi- sphere and North America and Greenland), JULES simulated similar GPP to MODIS, FLUXNET-MTE and CARDAMOM for the three biomes in Europe and the Extratrop- ical Southern Hemisphere (Figures 6a and d), with the exception of Northern Asia and North America and Greenland, where the model is either equal to or lower than all three datasets (Figures 6b and g). This is due to the inability of this version of JULES to accurately simulate GPP in boreal regions where permafrost exists.

[New comment] Can more evidence be provided as to why permafrost would explain these differences? I can see how lack of frozen soils in JULES might create more water available for plants and increase GPP, but this is opposite to what was found in the study. It looks like most of the bias is in the shrub-dominated regions (fig 6). So the problem could be a lack of appropriate PFTs for these cold environments. In the response the authors state that a different land cover map is assumed in JULES, MODIS, and FLUXNET-MTE – what about this source for explaining some of the differences?

5. [My original comment] Also the meteorological dataset did not strongly change the results. However this is dependent on two things: 1) Maybe there were not large differences in climate between the data sets? IE: Page 15, Lines 2-6: Why are these differences in GPP occurring? Is the temperature and precipitation (or other variables) very different between the datasets in these regions? Are there other regions where the climate is very different, but the JULES simulations do not show dramatically different GPP? It would be good to provide some more information on the climates from the different driving data sets.

[New comment] I disagree with the conclusion that longwave radiation is a cause for differences between the model results with different datasets. Alton et al. (2007) found only 0.6% impact of LW radiation on simulated GPP, compared to 5% for SW radiation.  There is no physiological reason to expect a strong link between longwave radiation and GPP. Also I don't see justification for the following statement in the cited Alton et al. paper:

"However, since JULES is **more sensitive to downward longwave radiation and surface air temperature than precipitation when simulating GPP** (Alton et al., 2007), the main reason for differences in simulated GPP when JULES was driven with two different meteorological datasets is due to differences in downward longwave radiation fluxes and surface air temperatures."

Alton et al. (2007) used constant soil moisture stress and soil resistance terms and so did not represent sensitivity of model fluxes to the cumulative effect of precipitation (see their section 4.5). Aren't there clues to the reasons for difference between the meteorological datasets in the discussion of limiting factors for photosynthesis? The fact that the WFDEI-GPCC runs are more light-limited agrees with Fig. G5 in the PhD thesis cited – this dataset also has lower downwelling SW radiation than the Princeton data.

The authors state: "In general, precipitation in the WFDEI-GPCC dataset is higher than that of PRINCETON (Figures G.6b and d in Slevin (2016)) with surface air temperatures higher in PRINCETON (Figures G.6a and c in Slevin (2016))" (~Line 10, page 15). As the authors have pointed out elsewhere in the paper and references cited within, JULES tends to decrease GPP with higher temperatures in the tropics, and moisture availability can directly impact the GPP through the soil moisture stress function. Therefore, both of these biases (higher precipitation and lower temperatures in GPCC) could also explain the higher GPP simulated with WFDEI-GPCC.

**Other comments**
1. Is there a relationship between the high bias in the tropics (Fig. 2) and the high bias during DJFM in Fig. 3? If so, that would imply that GPP is too high in the tropics during the wet season, and could give some clues to the reason for the over-estimation.

2. The added description of driving data (beginning at the end of Page 3) and how it influences JULES GPP is helpful but I think there are a few points to clarify:
   - Relating to my above comment, I think it is misleading to say that downwelling shortwave *and longwave* radiation play an important role in the calculation of photosynthesis.
   - The light-limited rate is only a function of downwelling shortwave radiation
   - The soil moisture stress is definitely affected by precipitation but it is not part of the calculation, which the new sentence at the end of the paragraph seems to imply.

3. It's great that the model outputs have been made available. Is it possible to share other data used to drive JULES, for example the soil data and PFT distribution?

4. How were the biomes determined? Do the "forest", "grassland", and "shrub" biomes correspond to grid cells where these vegetation types are dominant?

5. In the zonal mean plot, there is a large low bias in JULES in the sub-tropics (15-30N), but this is not mentioned until the discussion. I think this should be mentioned earlier, especially because the low bias is apparently overwhelmed by the tropical high bias in the biome-scale plots for the Tropics (30S-30N). Also if the trees/shrubs/grasses for sub-tropics were included in Figure 4, this would back up the claim that a drought-deciduous tree or shrub PFT would help improve this large model bias. This is just a suggestion if the authors agree it would be useful to add this distinction.

6. Discussion: There is still some repetition in the discussion, I would advise additional proof-reading to see where repetition can be removed and similar threads of the discussion can be joined together. I think there should be more of a link between the discussion of the transport vs rubisco limited rates and the biases found in this study.

---

## Author Response (AR2)

**Response to Anonymous Referee #2**

We thank the referee for providing a further review of the manuscript and agree that the suggested changes and clarifications improve it. We have made the changes outlined below in the revised manuscript. Each item starts with the reviewer's comment (in bold) followed by the changes to the manuscript. The text in blue is a re-written or new paragraph/sentence which has been added to the manuscript. The page and line numbers of where changes have been made to the updated manuscript are included at the end of each reply.

**Major Comments**

Overall the authors have addressed most of the points I raised in my initial review. I had two major comments which the authors responded to in detail. I am satisfied with most responses but below are two further questions based on these responses.

1. **(My original comment) (Discussion of figure 6): Why are the results for the extratropics the only ones discussed? I think much more could be said here - instead of just listing the differences it would be better to provide some more evaluation.**
   **(Author Response) Yes we found that JULES performs reasonably well in the extratropics (Europe, Northern Asia, North America and Greenland and the Extratropical Southern Hemisphere), with the exception of Northern Asia and North America and Greenland, where the model is either equal to or lower than all three datasets. This may be due to the inability of this version of JULES to accurately simulate GPP in boreal regions where permafrost exists. It may also due to a different land cover map being used by JULES, MODIS and FLUXNET-MTE. The following paragraph were added to section 4.2 (Page 14, lines 4–8).**
   **In the four extratropical regions (Europe, Northern Asia, Extratropical Southern Hemi- sphere and North America and Greenland), JULES simulated similar GPP to MODIS, FLUXNET-MTE and CARDAMOM for the three biomes in Europe and the Extratropical Southern Hemisphere (Figures 6a and d), with the exception of Northern Asia and North America and Greenland, where the model is either equal to or lower than all three datasets (Figures 6b and g). This is due to the inability of this version of JULES to accurately simulate GPP in boreal regions where permafrost exists.**
   **(New comment) Can more evidence be provided as to why permafrost would explain these differences? I can see how lack of frozen soils in JULES might create more water available for plants and increase GPP, but this is opposite to what was found in the study. It looks like most of the bias is in the shrub-dominated regions (fig 6). So the problem could be a lack of appropriate PFTs for these cold environments. In the response the authors state that a different land cover map is assumed in JULES, MODIS, and FLUXNET-MTE – what about this source for explaining some of the differences?**
   Yes it makes more sense that a lack of frozen soils in the permafrost regions in JULES means there is more water available for plants and therefore leads to increases in GPP. This increase in GPP can be seen in Figure 5b (Figures 5 and 6 have been swapped to ensure they are referred to in the correct order in the manuscript). As suggested, two

possible solutions to fixing this positive bias include the addition of a shrub PFT more suited to cold environments and the different land cover maps used by JULES, MODIS, and FLUXNET-MTE. The final sentence of Section 4.2 was removed and replaced with the following paragraph (Page 14, lines 26–33).

One possibility for the higher GPP estimates in Northern Asia and North America and Greenland may be due to a different land cover map being used by the observation-based datasets (MODIS and FLUXNET-MTE) and JULES. The PFT fractions specified in the land cover map used by JULES were used to calculate the GPP estimates for the forests, grasslands and shrubs biomes. So for a particular gridbox, the land cover map of JULES may specify a shrub, but in the land cover map used by MODIS or FLUXNET-MTE, it may be a needleleaf tree. Another possibility is that shrubs in these northern regions have adapted to the cold environment and the lower surface air temperatures have a lesser impact on photosynthesis than they do on shrubs in warmer climates. The addition of a shrub PFT to JULES which is adapted to cold climates may improve GPP estimates in these regions.

5. **(My original comment) Also the meteorological dataset did not strongly change the results. However this is dependent on two things: 1) Maybe there were not large differences in climate between the data sets? IE: Page 15, Lines 2–6: Why are these differences in GPP occurring? Is the temperature and precipitation (or other variables) very different between the datasets in these regions? Are there other regions where the climate is very different, but the JULES simulations do not show dramatically different GPP? It would be good to provide some more information on the climates from the different driving data sets.**

   **(New comment) I disagree with the conclusion that longwave radiation is a cause for differences between the model results with different datasets. Alton et al. (2007) found only 0.6% impact of LW radiation on simulated GPP, compared to 5% for SW radiation. There is no physiological reason to expect a strong link between longwave radiation and GPP. Also I don't see justification for the following statement in the cited Alton et al. paper: "However, since JULES is more sensitive to downward longwave radiation and surface air temperature than precipitation when simulating GPP (Alton et al., 2007), the main reason for differences in simulated GPP when JULES was driven with two different meteorological datasets is due to differences in downward longwave radiation fluxes and surface air temperatures."**

   **Alton et al. (2007) used constant soil moisture stress and soil resistance terms and so did not represent sensitivity of model fluxes to the cumulative effect of precipitation (see their section 4.5). Aren't there clues to the reasons for difference between the meteorological datasets in the discussion of limiting factors for photosynthesis? The fact that the WFDEI-GPCC runs are more light-limited agrees with Fig. G5 in the PhD thesis cited – this dataset also has lower downwelling SW radiation than the Princeton data.**

   **The authors state: "In general, precipitation in the WFDEI-GPCC dataset is higher than that of PRINCETON (Figures G.6b and d in Slevin (2016)) with surface air temperatures higher in PRINCETON (Figures G.6a and c in Slevin (2016))" ( Line 10, page 15). As the authors have pointed out elsewhere in the paper and references cited within, JULES tends to decrease GPP with higher temperatures in the tropics, and moisture availability can directly**

impact the GPP through the soil moisture stress function. Therefore, both of these biases (higher precipitation and lower temperatures in GPCC) could also explain the higher GPP simulated with WFDEI-GPCC.

Yes you are right in saying that the downwelling shortwave radiation is more important than the downwelling longwave radiation when calculating photosynthesis in JULES. The statement

"However, since JULES is more sensitive to downward longwave radiation and surface air temperature than precipitation when simulating GPP (Alton et al., 2007), the main reason for differences in simulated GPP when JULES was driven with two different meteorological datasets is due to differences in downward longwave radiation fluxes and surface air temperatures."

has been removed. The WFDEI-GPCC driven runs are more light-limited than those driven with PRINCETON (Figure 5.27 in Slevin (2016)) as the WFDEI-GPCC dataset has lower downwelling shortwave radiation than PRINCETON in general (Figures G.5a and c in Slevin (2016)). In the discussion of the importance of the various meteorological drivers in simulating GPP in different regions, section 4.4 was re-written (Page 15–16). In the sensitivity study performed in Slevin (2016), JULES simulated higher GPP with decreasing surface air temperatures in the tropics. The lower air temperatures and the higher precipitation in the WFDEI-GPCC dataset are the likely reason for the higher GPP simulated by JULES in the tropical regions. The following paragraph was added to section 4.4 (Page 15, lines 25–30).

The higher simulated GPP between 5°N–5°S in the Amazonian, African and South-East Asian tropics when JULES was driven with WFDEI-GPCC is due to the lower surface air temperatures (Figures G.6a and c in Slevin (2016)) and higher precipitation (Figures G.6b and d in Slevin (2016)) in the WFDEI-GPCC dataset(Slevin, 2016). In extratropical regions, such as northern Eurasia (above 60°N), there are differences in the meteorological datasets with slightly higher downward shortwave radiation fluxes and surface air temperatures in the PRINCETON dataset with little difference between the JULES simulations driven with either WFDEI-GPCC or PRINCETON in this region (Figure **??**).

**Other Comments**

1. **Is there a relationship between the high bias in the tropics (Fig. 2) and the high bias during DJFM in Fig. 3? If so, that would imply that GPP is too high in the tropics during the wet season, and could give some clues to the reason for the over-estimation.**

   We have suggested two possible reasons for the higher simulated GPP in the tropics. The following paragraph was added to Section 4.2 (Page 14, lines 11–19).

   There are two possible reasons for the larger simulated GPP in the tropics at 30°S–15°N. Firstly, the high bias in the tropics during December, January, February and March (Figures 2e and 3b) imply that JULES GPP is overestimated in the tropical wet season. The lower air temperatures and higher soil moisture availability during the wet season leads to higher simulated GPP in these regions. Secondly, the higher GPP in the tropics is due to the incorrect simulation of GPP by the version of JULES (version 3.4.1) used in this study. In this version, the PFT used to represent tropical forests is the broadleaf tree, which is used to simulate GPP in both tropical and temperate regions. This means that the model parameters used for the broadleaf tree PFT may not be suitable for simulating GPP in the tropics. The addition of extra PFTs more suited to tropical regions, such as tropical

broadleaf evergreen (in version 4.2) and a drought-deciduous PFT, and a phenology model which simulates LAI in tropical regions would both improve GPP simulations.

2. **The added description of driving data (beginning at the end of Page 3) and how it influences JULES GPP is helpful but I think there are a few points to clarify:**

   – **Relating to my above comment, I think it is misleading to say that downwelling shortwave and longwave radiation play an important role in the calculation of photosynthesis.**

   – **The light-limited rate is only a function of downwelling shortwave radiation.**

   – **The soil moisture stress is definitely affected by precipitation but it is not part of the calculation, which the new sentence at the end of the paragraph seems to imply.**

   Yes you are correct. The downwelling shortwave radiation plays a more important role than the downwelling longwave radiation in the calculation of photosynthesis. The paragraph describing how JULES calculates photosynthesis has been updated (Page 3, line 23–Page 4, line 2). The sentence regarding the link between soil moisture stress and photosynthesis has been re-written to show how it is connected to the calculation of leaf-level photosynthesis (Page 3, line 31–Page 4, line 2).

   Soil moisture stress is taken into account when calculating leaf-level photosynthesis by multiplying the potential leaf-level photosynthesis by a soil moisture factor (determined using mean soil moisture concentration in the root zone).

3. **It's great that the model outputs have been made available. Is it possible to share other data used to drive JULES, for example the soil data and PFT distribution?**
   The ancillary data (soil, vegetation, etc.) used for the simulations in this study have now been deposited on the University of Edinburgh DataShare service (`http://dx.doi.org/10.7488/ds/1995`). The dataset includes the ancillary data (soil, vegetation, etc) at the 3 different horizontal spatial resolutions ($0.5° \times 0.5°$, $1° \times 1°$ and $2° \times 2°$) and the annual $CO_2$ concentrations (parts per million by volume) required by JULES. Further information on the dataset's metadata can be found by clicking on the Show full item record link on the left hand side of the dataset's DataShare page. The doi for the ancillary data has been added to the Code and/or data availability section (Page 18, lines 26–27).

4. **How were the biomes determined? Do the "forest", "grassland", and "shrub" biomes correspond to grid cells where these vegetation types are dominant?**
   The biomes were determined using the PFT fractions from the land cover map used by JULES. Originally, the Forest, Grassland and Shrub biomes were determined by summing the PFT fractions in the land cover map for the broadleaf and needleleaf tree surface types, the C3 and C4 surface types and the shrub surface type, respectively. However, this is not accurate since it assumes that GPP can be attributed to non-vegetation land cover types. The Forest, Grassland and Shrub biome fractions are now further divided by the sum of the fractions of the 5 PFTs to ensure that the gridbox GPP has been calculated by the appropriate vegetation type. The following text was added to section 2.4 (Page 7, lines 15–18).

The Forest, Grassland and Shrub biomes were determined by summing the PFT fractions in the land cover map for the broadleaf and needleleaf tree surface types, the C3 and C4 surface types and the shrub surface type, respectively, and dividing each by the sum of the fractions of the 5 PFTs in order to exclude the non-vegetation land cover types.

This also means that the values of GPP for the 3 biomes used in the results and discussion sections have been updated (Pages 8–16). Figures 4 and 5 have also been updated.

5. **In the zonal mean plot, there is a large low bias in JULES in the sub-tropics (15-30N), but this is not mentioned until the discussion. I think this should be mentioned earlier, especially because the low bias is apparently overwhelmed by the tropical high bias in the biome-scale plots for the Tropics (30S-30N). Also if the trees/shrubs/grasses for sub-tropics were included in Figure 4, this would back up the claim that a drought-deciduous tree or shrub PFT would help improve this large model bias. This is just a suggestion if the authors agree it would be useful to add this distinction.**

The large negative bias in JULES in the subtropics at 15°N–30°N is now mentioned earlier in the results section (Sections 3.1 (Page 9, lines 1–4) and 3.3 (Page 10, lines 3–5)). However, reasons for the large negative bias are still discussed in more detail in section 4.1. Figure 4 has been modified to include the total (summed over 10 years) model simulated (JULES-WFDEI-GPCC, JULES-WFDEI-CRU and CARDAMOM), observation-based (FLUXNET-MTE and MODIS) GPP fluxes for forests, grasses and shrubs further subdivided into GPP for the tropics at 30°S–15°N (subfigure d) and Mexico (subfigure e). The colour bar for Figure 8 (Figure 7 has now become Figure 8) was updated to include both positive and negative changes in GPP. Differences in the monthly mean JULES and MODIS LAI was added to Figure 8 as subfigure d. The discussion of this negative bias in JULES has been re-written to include new insight from Figures 4d, 4e and 8d. The following text was added to section 4.1 (Page 12, line 15–Page 13, line 2).

JULES simulates lower GPP than MODIS, FLUXNET-MTE and CARDAMOM at 15°N–30°N (Figures 6 and 8). This large negative bias in JULES was due to the incorrect simulation of GPP in subtropical regions such as Mexico and southern China (Figures 8a, b, and c). The total annual MODIS and FLUXNET-MTE GPP estimates for 2001–2010 are higher than that simulated by JULES by 1% and 7%, respectively, for Mexico, with CARDAMOM GPP estimates for the same period being lower than JULES GPP by 6%. The tropical GPP fluxes for forests, grasslands and shrubs were further subdivided into two regions; (1) the tropics at 30°S–15°N (Figure 4d) and (2) Mexico (Figure 4e). Trends in JULES GPP were similar for the two tropical regions at 30°S–30°N and 30°S–15°N with positive biases in forest and grassland GPP (Figures 4b and d). GPP in Mexico was similar for forests and grasslands with differences in shrub GPP (Figure 4e). The negative bias in JULES GPP in the subtropics is due to low LAI simulated by the model compared to MODIS (Figure 5.10 in Slevin (2016); Figure 8d). MODIS LAI is used as input when generating the MODIS, FLUXNET-MTE and CARDAMOM GPP estimates.

One of the major vegetation types in the subtropics is drought-deciduous plants (drought-deciduous plants lose their leaves during the dry season or periods of dryness as opposed to temperate deciduous plants which lose their leaves during periods of cold weather) and JULES does not contain this PFT. Drought-deciduous plants can be found in the seasonally dry tropical forests of Mexico, Central America, northwestern South America and southern China. The implementation of a drought-deciduous shrub PFT would help improve simulated GPP in these regions. In JULES, phenology is updated once per day by multiplying the annual maximum LAI by a scaling factor, which is calculated using

temperature-dependent leaf turnover rates. Leaf turnover rates are a function of surface air temperature and increase when the temperature drops below a certain value (this varies depending on the PFT). While this is suitable for deciduous broadleaf forests in temperate regions, such as Northern Europe, it will lead to inaccurate modelled LAI for drought-deciduous forests. Instead of modifying modelled LAI using a temperature-derived scaling factor, the scaling factor could be calculated by using periods of dryness as the controlling factor.

6. **Discussion: There is still some repetition in the discussion, I would advise additional proofreading to see where repetition can be removed and similar threads of the discussion can be joined together. I think there should be more of a link between the discussion of the transport vs rubisco limited rates and the biases found in this study.**

   Sections 4.1, 4.2 and 4.4 in the discussion were proofread to remove repetition (Pages 11–16). The text in this section was modified in order to provide more of a link between the discussion of the transport vs rubisco limited rates and the biases found in this study (Page 13, lines 3–18; Page 16, lines 3–16).

When JULES was driven with the PRINCETON dataset, it was found that simulated photosynthesis was mostly Rubisco-limited (Figure Figures 5.25 and 5.6 in Slevin (2016)). A similar trend was found This is a similar result to when JULES was driven with the WFDEI-GPCC dataset(Figure 5.6 in Slevin (2016)). Similar trends in transport limitation were found with the JULES-PRINCETON model simulation, though the number of model gridboxes in which transport limitation dominated was less than that for the JULES-WFDEI-GPCC-1degree model simulation (Figures 5.25 and 5.28 in Slevin (2016)). When comparing the . When model gridbox fractions for the JULES-WFDEI-GPCC-1degree and JULES-PRINCETON 
[revised manuscript text omitted]

---

## Author Response (AR3)

**Response to the Topical Editor**

We thank the topical editor for providing a further review of the manuscript and agree that the suggested minor changes and clarifications improve it. We have made the changes outlined below in the revised manuscript. Each item starts with the editor's comment (in bold) followed by the changes to the manuscript. The text in blue is a re-written or new paragraph/sentence which has been added to the manuscript. The page and line numbers of where changes have been made to the updated manuscript are included at the end of each reply.

**Minor Corrections**

1. **P1L7 "JULES was found to simulate interannual variability (IAV) at global scales." Rephrase this sentence more specifically.**
   This sentence was re-written as (Page 1, lines 7–9)
   "JULES was able to simulate the standard deviation of monthly GPP fluxes compared to CARDAMOM and the observation-based estimates at global scales."

2. **P7L22 "CDO". This manuscript employed too much abbreviations. As the CDO is an infrequent term, replace it with full words.**
   CDO has been replaced with its full meaning (Page 7, lines 26–27).

3. **P15L20 "(Table2," Missing a closing parenthesis.**
   The missing closing parenthesis has now been added (Page 15, line 23).

4. **P17L9 "because of a lack of drought-deciduous PFTs in JULES"**
   **P18L14 "due to lack of drought-deciduous PFTs in JULES"**
   **Avoid to mention it in a predicable way. Lack of drought-deciduous PFTs would NOT be the only possible explanation for difference between simulation and observation.**
   Yes, you are correct. A lack of drought-deciduous PFTs would NOT be the only possible explanation for differences between JULES and the observations in the subtropics. The sentence
   "Differences in GPP between JULES and the benchmarking datasets (FLUXNET-MTE, MODIS and CARDAMOM) at 15°N–30°N is due to higher FLUXNET-MTE, MODIS and CARDAMOM GPP in regions such as Mexico and southern China because of a lack of drought-deciduous PFTs in JULES."

   has been changed to (Page 17, lines 7–9)

   "Differences in GPP between JULES and the benchmarking datasets (FLUXNET-MTE, MODIS and CARDAMOM) at 15°N–30°N is due to higher FLUXNET-MTE, MODIS and CARDAMOM GPP in regions such as Mexico and southern China. These differences may be due to a lack of drought-deciduous PFTs in JULES."

   and the sentence

   "In general, differences between JULES GPP and the benchmarking datasets (FLUXNET-MTE, MODIS and CARDAMOM) occur mostly in the tropics with differences at 15°N–30°N due to a lack of drought-deciduous PFTs in JULES."

has been changed to (Page 18, lines 13–14)

"In general, differences between JULES GPP and the benchmarking datasets (FLUXNET-MTE, MODIS and CARDAMOM) occur mostly in the tropics with differences at 15°N–30°N possibly due to a lack of drought-deciduous PFTs in JULES."

5. **P17L12 "The model is able to simulate" It would be better to replace this term with "The model can reasonably reproduce".**
   This sentence has been changed to (Page 17, lines 12–13)

[revised manuscript text omitted]